# ENERGY-CONSERVING EQUIVARIANT GNN FOR ELASTICITY OF LATTICE ARCHITECTED METAMATERIALS

**Ivan Grega**[1]*, **Ilyes Batatia**[1], **Gábor Csányi**[1], **Sri Karlapati**[1,2,†], **Vikram S. Deshpande**[1]

[1] Department of Engineering, University of Cambridge; [2] Amazon Science

## ABSTRACT

Lattices are architected metamaterials whose properties strongly depend on their geometrical design. The analogy between lattices and graphs enables the use of graph neural networks (GNNs) as a faster surrogate model compared to traditional methods such as finite element modelling. In this work, we generate a big dataset of structure-property relationships for strut-based lattices. The dataset is made available to the community which can fuel the development of methods anchored in physical principles for the fitting of fourth-order tensors. In addition, we present a higher-order GNN model trained on this dataset. The key features of the model are (i) *SE(3) equivariance*, and (ii) consistency with the thermodynamic law of *conservation of energy*. We compare the model to non-equivariant models based on a number of error metrics and demonstrate its benefits in terms of predictive performance and reduced training requirements. Finally, we demonstrate an example application of the model to an architected material design task. The methods which we developed are applicable to fourth-order tensors beyond elasticity such as piezo-optical tensor etc.

## 1 INTRODUCTION

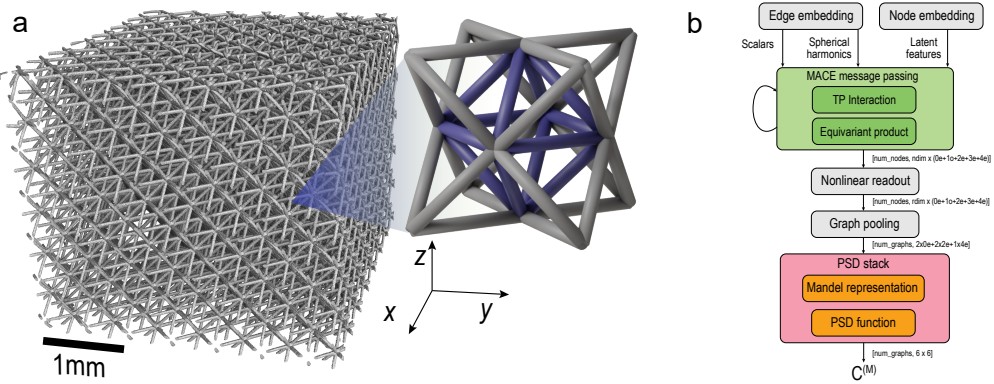

Figure 1: *(a)* X-ray CT scan of 3d-printed lattice. A computer model of the unit cell is shown as an inset. *(b)* Model schematic. The dimensionality of intermediary quantities is noted between layers using `e3nn` convention. We omit simple linear layers from the diagram for clarity.

A relatively new class of materials, *architected (meta-)materials*, emerged in the last century. (Fleck et al., 2010) These materials draw inspiration from nature, where many materials are light, yet strong, because of their porosity and microscopic architecture. As a subclass of architected materials, *lattices* are a collection of *struts* (edges) which are connected at *nodes*. See Figure 1a and Figure 5 in the Appendix. Lattices are especially mechanically efficient, offering a very high *specific stiffness* (stiffness divided by density). For instance, it is possible to make materials with the density of water and the strength of steel.

---

*`ig348@cam.ac.uk`    † Work done outside of Amazon Science through an informal collaboration.

The established tool for computational analysis of lattices is the *finite element* (FE) method, which is also the industry standard for other structures from buildings to cars and airplanes. There are a number of principles which the FE solution satisfies (subject to a suitable PDE and constitutive model). First, force equilibrium is satisfied at all nodes and the computed displacements are compatible. Second, the strain energy under any deformation is nonnegative as required by energy conservation. Third, results are equivariant to rigid body transformations: rotating the lattice does not change its fundamental properties; they rotate accordingly.

Although the FE method is robust and physically grounded, its high computational cost can be prohibitive: for example, if each unit cell of size 1cm is discretized into 100 elements, a wing structure of $\sim$ 20m length would require $n \sim 10^9$ elements.

Machine learning methods have been used to overcome the computational cost of FE methods. Indurkar et al. (2022) employed message-passing GNN to classify lattices based on their mechanical response. Karapiperis & Kochmann (2023) used GNN to predict the crack path in disordered lattices. Xue et al. (2023) build a GNN to learn the non-linear dynamics of mechanical metamaterials. Maurizi et al. (2022) use GNN to predict the mechanical response of composites and lattices. Meyer et al. (2022) have presented a GNN framework to predict the stiffness tensor of *shell-based* lattices. Machine learning methods have also been used to do inverse design of materials. Kumar et al. (2020) couple inverse and forward models to design spinodoid materials with orthotropic symmetry. Bastek et al. (2022) use a similar models for strut-based lattices with fully tailorable 3D anisotropic stiffness. Zheng et al. (2023) build a VAE model for generation of lattices with up to 27 nodes and cubic symmetry.

While these machine learning models offer a much higher speed than FE, they *lack grounding in physical principles.* This might not be an issue when the application is restricted to a particular lattice symmetry class and when the model is deployed for data that are close to the training distribution. However, models without encoded equivariance and energy conservation principles could fail dramatically if deployed to out-of-distribution lattice topologies – the predictions for the same lattice at different orientations might not be self-consistent, and negative deformation energy could be predicted, implying the ability to extract energy from passive material.

In this work, we rely on the equivariant methods which have been introduced by the computational chemistry community. (Thomas et al., 2018; Batzner et al., 2022; Batatia et al., 2022) As key contributions:

- We introduce a **new task** into the ML community and provide a real-world **dataset** which can be used by researchers in the future to improve higher order physics-focused models.

- We present one such model – the first **equivariant** model trained for prediction of the fourth-order elasticity tensor whose predictions are always **energy conserving** (consistent with the laws of physics).

- We benchmark the model against non-equivariant models and show the benefits of key model components and training strategies.

## 2 BACKGROUND

### 2.1 EQUIVARIANCE

In the domain of physical sciences, invariance or equivariance under some physical transformations is an important property. For example, in chemistry, the energy of a molecule needs to be the same regardless of the coordinate system chosen to represent the coordinates of the atoms. In our modeling of lattices, model predictions need to satisfy similar rules. Let $\mathcal{L}$ represent a lattice that has attributes of the following types: *scalars* (e.g. edge lengths $L$), *vectors* (e.g. edge directions $\boldsymbol{v} = v_i$), *tensors* (e.g. 4$^{\text{th}}$ order stiffness tensor $\mathbf{C} = C_{ijkl}$).

In the discussion of equivariance, we focus on two main actions: rigid body *rotation* and *translation*. Let $Q(.\,;\boldsymbol{R})$ represent the rotation given by the rotation matrix $\boldsymbol{R}$ applied on the object, which is the first argument of the function. The following analytical transformation rules apply for scalars $L$,

vectors $\boldsymbol{v}$, and higher order tensors $\mathbf{K}$:

$$\hat{L} = Q(l; \boldsymbol{R}) = L$$
$$\hat{v}_i = Q(\boldsymbol{v}; \boldsymbol{R}) = R_{ij} v_j$$
$$\hat{K}_{ijk...} = Q(\mathbf{K}; \boldsymbol{R}) = R_{ia} R_{jb} R_{kc} ... K_{abc...}$$

All these objects are *invariant* to rigid body translation.

The notation $Q(\mathcal{L}; \boldsymbol{R})$ represents the rotation of the lattice $\mathcal{L}$, which implies the rotation of all attributes of the lattice according to the aforementioned transformation rules. Translation of the lattice $T(\mathcal{L}; \boldsymbol{t})$ simply means displacing the nodal positions by the vector $\boldsymbol{t}$: $\boldsymbol{x} \leftarrow \boldsymbol{x} + \boldsymbol{t}$.

Our task is to predict the stiffness tensor $\mathbf{C}$ for the lattice $\mathcal{L}$. The prediction of model $\mathcal{M}$ is $\mathcal{M}(\mathcal{L})$. The equivariance requirement is then

$$\mathcal{M}(T(\mathcal{L}; \boldsymbol{t})) = \mathcal{M}(\mathcal{L}) \quad \forall \boldsymbol{t}$$
$$Q(\mathcal{M}(\mathcal{L}); \boldsymbol{R}) = \mathcal{M}(Q(\mathcal{L}; \boldsymbol{R})) \quad \forall \boldsymbol{R}$$

## 2.2 EUCLIDEAN EQUIVARIANT MESSAGE PASSING NEURAL NETWORKS

**Message Passing Neural Networks** Euclidean Equivariant Message Passing Neural Networks (MPNNs) (Liao & Smidt, 2023; Thomas et al., 2018; Weiler et al., 2018; Kondor et al., 2018; Batzner et al., 2022; Brandstetter et al., 2022; Batatia et al., 2022; Satorras et al., 2021) are graph neural networks that are equivariant to rotations and translations. MPNNs map a graph $\mathcal{G}$ with labels called states $\sigma_i$ on each node $i$, to a target $y$. At each layer $t$, MPNNs operate in four successive steps, the *edge embedding*, the *pooling*, the *update* and the *readout*,

$$\boldsymbol{m}_i^{(t)} = \bigoplus_{j \in \mathcal{N}(i)} M_t(\sigma_i^{(t)}, \sigma_j^{(t)}), \quad \boldsymbol{h}_i^{(t+1)} = U_t(\sigma_i^{(t)}, \boldsymbol{m}_i^{(t)}), \quad y = \mathcal{R}_t(\{\sigma_i^{(t)}\}_{i,t}) \tag{1}$$

where $M_t$ is the edge embedding function, $\bigoplus_{j \in \mathcal{N}(i)}$ is the pooling operation (usually just a sum) over the neighborhood of the node $i$, $\mathcal{N}(i)$. $U_t$ is the update function. These steps are repeated $T$ times. Finally, the readout $\mathcal{R}_t$ maps the states to the target quantity.

**Equivariant MPNNs** Most Euclidean MPNNs expand their internal features in a spherical basis. Node features carry an index $lm$ specifying the order of the basis expansion.

$$h_{i,lm}^{(t)}(R \cdot (r_1, \ldots, r_N)) = \sum_{m'} D_{m',m}^l(R) h_{i,lm'}^{(t)}(r_1, \ldots, r_N) \tag{2}$$

with $D_{m',m}^l(R)$ the Wigner-D matrices corresponding to the action of the rotation group on the spherical basis. Therefore, this $lm$ index is carried over to all internal features of the model.

**Higher order MPNNs** In full generality, the message can be a simultaneous function of all neighboring atoms of the central atoms $i$. Therefore, one can expand the message in a many-body expansion of the states,

$$\boldsymbol{m}_i^{(t)} = \sum_j \boldsymbol{u}_1\left(\sigma_i^{(t)}; \sigma_j^{(t)}\right) + \sum_{j_1, j_2} \boldsymbol{u}_2\left(\sigma_i^{(t)}; \sigma_{j_1}^{(t)}, \sigma_{j_2}^{(t)}\right) + \cdots + \sum_{j_1, \ldots, j_\nu} \boldsymbol{u}_\nu\left(\sigma_i^{(t)}; \sigma_{j_1}^{(t)}, \ldots, \sigma_{j_\nu}^{(t)}\right) \tag{3}$$

The number of simultaneous dependency is called the body-order. MPNN potentials were shown to increase the body order of messages by stacking layers. An alternative route is to include higher-order terms in the message construction. The MACE (Batatia et al., 2022) architecture, on which we will be building in this work, introduced a systematic way to efficiently approximate equivariant messages of an ordered arbitrary body.

## 2.3 SOLID MECHANICS

We consider lattices as infinite periodic tessellations of a *unit cell*. The resulting *metamaterial* can be characterized by macroscopic (homogenized) properties. The key variables in solid mechanics

under the assumption of small deformations are *stress*, $\boldsymbol{\sigma} = \sigma_{ij}$, which is a measure of force, and strain $\boldsymbol{\epsilon} = \epsilon_{kl}$, which is a measure of deformation. Both stress and strain are symmetric $3 \times 3$ second order tensors ($\sigma_{ij} = \sigma_{ji}$, $\epsilon_{kl} = \epsilon_{lk}$).

A third key component in solving a solid mechanics problem is the *constitutive law*, which relates stress and strain. Linear elasticity postulates that $\sigma_{ij} = C_{ijkl}\epsilon_{kl}$ where $\mathbf{C} = C_{ijkl}$ is the fourth-order *stiffness* tensor.

*Deformation energy* $\psi$ under deformation $\boldsymbol{\epsilon}$ is given by the following tensor contraction. Importantly, thermodynamic laws prescribe non-negative deformation energy $\psi \geq 0$ for any admissible deformation $\boldsymbol{\epsilon}$:

$$\psi = \frac{1}{2}\sigma_{ij}\epsilon_{ij} = \frac{1}{2}\epsilon_{ij}C_{ijkl}\epsilon_{kl} \quad \geq 0 \qquad \forall \boldsymbol{\epsilon} \tag{4}$$

Since this has to be true for all strains $\boldsymbol{\epsilon}$, all eigenvalues of the stiffness tensor $\mathbf{C}$ must be non-negative. *The stiffness tensor must be positive semi-definite.*

The $4^{\text{th}}$ order stiffness tensor $C_{ijkl}$ is a $3 \times 3 \times 3 \times 3$ tensor. While a tensor with such dimensionality could have up to 81 components, it can be easily shown that the tensor has only 21 independent components, because it possesses both minor and major symmetries (Section A.1):

$$C_{ijkl} = C_{jikl} = C_{ijlk} = C_{klij}$$

## 3 RELATED WORK

**Finite element (FE) modelling**   The gold standard in computational methods in mechanics has been finite element modelling. In the FE framework, the properties of constituent material (e.g. steel) are known, and FE is used to calculate the structural response. The structure has degrees of freedom $u_j$, and it is loaded by external forces $f_i$. The first step to solving a FE problem is to assemble stiffness matrix $K_{ij}$, which relates displacements and forces: $f_i = K_{ij}u_j$. The next step is to solve this matrix equation for $\boldsymbol{u}$ (usually by LU factorization). If properties of the material are known, FE provides very accurate predictions of the overall structural response. However, the computational complexity of the matrix inversion (or LU factorization) can be very high.

**Dataset**   Lumpe & Stankovic (2021) explored the property space of a large dataset of mechanical lattices. The dataset which they used and made available comes from two crystallographic databases (Ramsden et al., 2009; O'Keeffe et al., 2008). The assembled dataset includes nodal positions, edge connectivity, crystal constants, and some elastic properties (Young's moduli, shear moduli and Poisson's ratios in the three principal directions). We use the crystallographic structures from this dataset as a basis for our dataset here.

**Crystal Graph Convolutions (CGC) and modified CGC (mCGC)**   To our knowledge, the only existing GNN model used to predict the *stiffness tensor* of architected materials is due to Meyer et al. (2022). Instead of beam-based lattices, the authors fitted the stiffness tensor of shell-based lattices. Their model is not equivariant. They use a form of data augmentation whereby each lattice is rotated 90° around the $x-$, $y-$ and $z-$ axis, and mirrored about the $x-y$, $y-z$, and $x-z$ planes. This increased the size of the training dataset 7-fold. The loss used in training is component-wise smooth_L1 on the 21 independent components of the stiffness tensor.

**NNConv for 3d lattices**   Ross & Hambleton (2020) use GNN to model cubic lattices with 48 rotational and reflectional symmetries. Their model is based on the NNConv layer, which was introduced by Gilmer et al. (2017). In the NNConv model, messages between nodes are formed as a matrix-vector product, where the entries of the matrix are not constant but rather depend on the features of the edge connecting the two nodes.

**E(3)-Equivariant Message Passing Neural Networks**   Equivariant Message Passing Neural Networks (MPNNs) Anderson et al. (2019); Thomas et al. (2018); Brandstetter et al. (2022); Batzner et al. (2022); Batatia et al. (2022) are a class of GNNs that respect Euclidean symmetries (rotations, reflections, and translations). Messages are usually expanded in a spherical basis, and depending on the order of expansion, not only vectors but also higher-order features such as tensors can be

passed between layers. Equivariant MPNNs have emerged as a powerful architecture for learning on geometric point clouds.

**Methods for enforcing positive (semi-)definiteness**   Jekel et al. (2022) review a number of methods that can be used to ensure that the output of a model is positive semi-definite. These include methods based on Cholesky factorization by (Xu et al., 2021; Van 't Sant et al., 2023) and methods based on eigenvalue decomposition. Note that these methods cannot be used in our framework because the eigenvalue decomposition often has unstable gradients and assembling the matrix by Cholesky factorization is not SO(3) equivariant.

# 4  METHODS

## 4.1  DATASET

We created a dataset on the basis of the dataset from Lumpe & Stankovic (2021). We process the dataset to fix or avoid problematic lattices which reduces the dataset size to 8954 base lattices. We augment the dataset by introducing nodal perturbations: e.g. for perturbation level 0.1, each node of a lattice is displaced from the original position by distance 0.1 in a random direction. After the new perturbed lattice is obtained, its elastic properties have to be computed using FE analysis. Nodal perturbations are applied at levels 0.01, 0.02, 0.03, 0.04, 0.05, 0.07, 0.1. At each level, we formed 10 distinct realizations of nodal perturbations. (Perturbations could only be applied to lattices which have at least 2 fundamental nodes.) This enlarged the entire database to $635\,454$ distinct geometries. For each geometry, FE analysis was run at 3 relative densities (strut thicknesses).

For the machine learning tasks in this paper, we selected from *base lattices*: (i) 7000 training base lattices, and (ii) 1296 validation/test base lattices. This split ensures that we do not have similar perturbations of the same lattice in both training and test sets.

See the Appendix for the various compositions of the training dataset. Validation and test sets are fixed for all training runs. Validation set consists of the 1296 lattices without any perturbations. Test set consists of 3 realizations of nodal perturbations at level 0.1 for the 1296 lattices. Thus, testing is done on OOD data.

## 4.2  ARCHITECTURE

The diagram in Figure 1b depicts the architecture of the model. Further details are explained in the Appendix. We highlight the main components here. The model relies on the MACE architecture for message passing which was adapted with minor changes. In particular, we used Gaussian embedding of edge scalars and all node features were initialised as ones and expanded using a linear layer. We modified the nonlinear readout to enable the processing of higher order tensors.

The significant contribution of this work is the positive semi-definite (PSD) stack. As a first step, the *fourth-order tensor* is transformed to Cartesian basis and then represented in Mandel notation as a *matrix*. Subsequently, a suitable PSD function is applied to the matrix, which enforces its positive semi-definiteness. This ensures energy conservation which was the key requirement of this work. The following section provides further details about the PSD stack.

## 4.3  MANDEL REPRESENTATION AND PSD LAYER

A fourth-order Cartesian tensor with major and minor symmetries $C_{ijkl} = C_{ijlk} = C_{jikl} = C_{klij}$ has 21 independent components. Suppose this fourth-order tensor is a map between second-order stress and strain:

$$\sigma_{ij} = C_{ijkl}\epsilon_{kl}$$

In Mandel notation, the second-order tensors can be written as 6-component vectors, and the fourth-order tensor **C** can be represented as a $6 \times 6$ symmetric matrix:

$$\boldsymbol{\sigma}^{(M)} = \left[\sigma_{11}, \sigma_{22}, \sigma_{33}, \sqrt{2}\sigma_{23}, \sqrt{2}\sigma_{13}, \sqrt{2}\sigma_{12}\right]^T \qquad \boldsymbol{\epsilon}^{(M)} = \left[\epsilon_{11}, \epsilon_{22}, \epsilon_{33}, \sqrt{2}\epsilon_{23}, \sqrt{2}\epsilon_{13}, \sqrt{2}\epsilon_{12}\right]^T$$

$$\boldsymbol{C}^{(M)} = \begin{bmatrix} C_{1111} & C_{1122} & C_{1133} & \sqrt{2}C_{1123} & \sqrt{2}C_{1113} & \sqrt{2}C_{1112} \\ C_{2211} & C_{2222} & C_{2233} & \sqrt{2}C_{2223} & \sqrt{2}C_{2213} & \sqrt{2}C_{2212} \\ C_{3311} & C_{3322} & C_{3333} & \sqrt{2}C_{3323} & \sqrt{2}C_{3313} & \sqrt{2}C_{3312} \\ \sqrt{2}C_{2311} & \sqrt{2}C_{2322} & \sqrt{2}C_{2333} & 2C_{2323} & 2C_{2313} & 2C_{2312} \\ \sqrt{2}C_{1311} & \sqrt{2}C_{1322} & \sqrt{2}C_{1333} & 2C_{1323} & 2C_{1312} & 2C_{1312} \\ \sqrt{2}C_{1211} & \sqrt{2}C_{1222} & \sqrt{2}C_{1233} & 2C_{1223} & 2C_{1213} & 2C_{1212} \end{bmatrix}$$

The energy conservation requirement can be rewritten as

$$\psi = \frac{1}{2}\sigma^{(M),T}\epsilon^{(M)} = \frac{1}{2}\epsilon^{(M),T}\boldsymbol{C}^{(M)}\epsilon^{(M)} \qquad \forall \epsilon^{(M)}.$$

Therefore, positive-definite fourth-order **C** is equivalent to positive-definite matrix $\boldsymbol{C}^{(M)}$.

We can apply various methods to enforce the positive definiteness of matrix $\boldsymbol{C}^{(M)}$. These include taking even powers of the matrix, $\boldsymbol{A}^2$ and $\boldsymbol{A}^4$, matrix exponential, and its truncated versions, $e^{\boldsymbol{A}}$, $(\boldsymbol{I} + \boldsymbol{A}/2)^2$, $(\boldsymbol{I} + \boldsymbol{A}/4)^4$.

*We prove in the Appendix that the PSD stack maintains equivariance of the framework.*

### 4.4 TRAINING AND EVALUATION DETAILS

Base CGC and mCGC models are trained according to the procedure described in ref Meyer et al. (2022). Model MACE is a plain version of MACE that is trained without data augmentation. Where "+tr" is added to the model name, it denotes that the model was trained using our training method as outlined below. The suffix "+ve" denotes a model which includes the *positive semi-definite layer* and was trained using our training method. In our training method, we use dynamic data augmentation, whereby every time a lattice is accessed from the dataset, it is returned at a different random orientation (and target stiffness is transformed accordingly). Further details about training including the equations for the various types of loss ($L_{\text{comp}}$, $L_{\text{dir}}$, $L_{\text{dir,rel}}$, $L_{\text{equiv}}$, $\lambda_{\%}^{-}$) are in the Appendix.

## 5 RESULTS

In this section we compare the performance of our model with other models and identify the key components of both the model and training procedures. We also show an example of a downstream application of the GNN model in a design task.

### 5.1 EQUIVARIANT MODELS OUTPERFORM NON-EQUIVARIANT MODELS

In Table 1, we show the performance of three main classes of models: CGC, NNConv, and MACE for dataset `limp` (find dataset details in the Appendix).[1] Crystal graph convolution (CGC) is based on works by Xie & Grossman (2018) and Meyer et al. (2022). In CGC, a *constant* learnt matrix multiplies node and edge features to create messages. Models NNConv and MACE are different in the nature of their message passing: the matrix which multiplies node features to obtain messages is a function of edge features. Details of all the models are explained in the appendix. Comparing the errors, it is evident that *the equivariant MACE model class achieves lowest errors by all metrics.*

We observe the following. (i) By adding data augmentation to CGC and NNConv models, models CGC+tr and NNConv+tr achieve substantially reduced stiffness-based errors ($L_{\text{comp}}$, $L_{\text{dir}}$, $L_{\text{dir,rel}}$), as well as equivariance loss, $L_{\text{equiv}}$. However, these models are more prone to predict negative eigenvalues ($\lambda_{\%}^{-}$). (ii) Adding data augmentation to equivariant MACE model leads to a much smaller improvement. The percentage of predicted negative eigenvalues, $\lambda_{\%}^{-}$, is not significantly affected. (iii) Models CGC+ve, NNConv+ve, and MACE+ve with encoded positive semi-definite output suffer in terms of increased stiffness-based errors. (iv) CGC-based models outperform NNConv-based models. NNConv models will therefore not be considered in the following sections.

---

[1] **The choice of lattice representation.** Meyer et al. (2022) combined message passing on primal and dual graph in model mCGC. We do not observe any performance gain from using dual graph representation on our data, therefore this model is omitted from the main discussion. We report more details in the Appendix (Table 8).

Table 1: Performance of the different models and training strategies

|  | CGC | CGC+tr | CGC+ve | NNConv | NNConv+tr | NNConv+ve | MACE | MACE+tr | MACE+ve |
|---|---|---|---|---|---|---|---|---|---|
| $L_{\text{comp}}$ | 8.37 | 4.47 | 5.47 | 8.99 | 5.57 | 7.07 | 3.88 | **3.47** | 3.61 |
| $L_{\text{dir}}$ | 8.77 | 5.23 | 5.86 | 9.65 | 6.90 | 8.08 | 4.17 | **4.11** | 4.21 |
| $L_{\text{dir,rel}}$ | 0.42 | 0.24 | 0.26 | 0.44 | 0.38 | 0.35 | 0.21 | **0.20** | 0.21 |
| $L_{\text{equiv}}$ | 0.33 | 0.15 | 0.17 | 0.39 | 0.25 | 0.22 | **0** | **0** | **0** |
| $\lambda_\%^-$ | 4 | 26 | 0 | 8 | 16 | 0 | 30 | 34 | **0** |

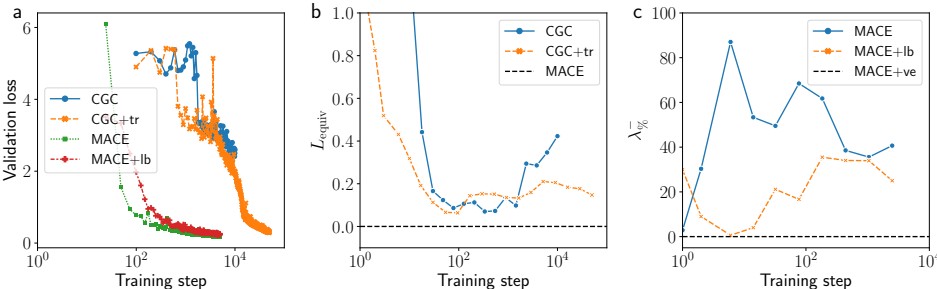

Figure 2: The evolution of *(a)* component loss, $L_{\text{comp}}$, *(b)* equivariance loss, $L_{\text{equiv}}$, and *(c)* percentage of negative eigenvalues, $\lambda_\%^-$, during training.

## 5.2 INDUCTIVE BIASES SUPERIOR TO OBSERVATION AND LEARNING BIASES

As outlined by Karniadakis et al. (2021), there are three conceptual pathways to embedding physics knowledge into machine learning models: *observation bias*, *learning bias* and *inductive bias*. Here we evaluate the efficacy of these biases from the viewpoint of equivariance and energy conservation.

Table 2: Performance of various bias types for the learning of equivariance and energy conservation

|  | Equivariance | | | | Energy conservation | | |
|---|---|---|---|---|---|---|---|
|  | none | observation | inductive | | none | learning | inductive |
|  | CGC | CGC+tr | MACE | | MACE | MACE+lb | MACE+ve |
| $L_{\text{equiv}}$ | 0.64 | 0.13 | 0 | $\lambda_\%^-$ | 30 | 29 | 0 |
| $L_{\text{comp}}$ | 9.31 | 4.47 | 3.88 | $L_{\text{comp}}$ | 3.88 | 3.63 | 3.61 |

**Equivariance learning** We achieve observation bias for equivariance by rotating the data that the model is trained on. As explained in section 4, models which end in "+tr" suffix were trained using data augmentation by rotation. Table 2 shows that the *equivariance error reduces dramatically* when we incorporate rotation augmentation of data. During training of model CGC in Table 2, all lattices were processed at a single orientation. Note that this is different from model CGC in Table 1 which was trained with 7-fold data augmentation as presented by Meyer et al. (2022). Figure 2b shows that the equivariance error reduces during training for both CGC and CGC+tr models. Not only is the final equivariance loss for model CGC+tr lower, but also the rate at which $L_{\text{equiv}}$ reduces is faster. It is instructive to note further that while validation loss keeps reducing during training (Figure 2a), the equivariance loss is not a monotonously decreasing function. Model MACE is equivariant by design, therefore equivariance loss, $L_{\text{equiv}}$, is zero both in Table 2 and Figure 2. Furthermore, the equivariant MACE model also has a lower component loss, $L_{\text{comp}}$.

**Energy conservation learning** A second physical principle that our model should be aligned with is the positive semi-definiteness of stiffness. We evaluate this for the MACE model in Table 2 and Figure 2c. The base MACE model trained on the data predicts negative eigenvalues for 30% of lattices. We attempt to introduce learning bias in model MACE+lb as follows. Loss is modified to include a penalty which is calculated from directional projections $c_q$ the of predicted stiffness tensor, $\tilde{\mathbf{C}}$, into 250 random directions $\boldsymbol{d}_q$: $c_q = \tilde{C}_{ijkl}d_{qi}d_{qj}d_{qk}d_{ql}$. The penalty is then computed as $k \times \text{relu}(-c_q)$ where $k$ is a suitably chosen multiplier. This penalty is added to loss during training.

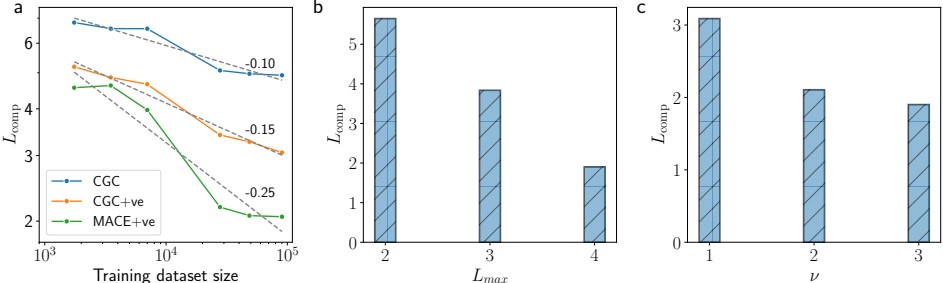

Figure 3: *(a)* Convergence with the amount of data for various model classes. *(b)* Sensitivity to the maximum 'frequency' $L_{max}$ and *(c)* correlation order $\nu$ for MACE model.

Table 3: Various methods for ensuring positive semi-definiteness for equivariant model

|  | $e^A$ | $A^2$ | $A^4$ | $(I + A/2)^2$ | $(I + A/4)^4$ |
|---|---|---|---|---|---|
| $L_{\text{comp}}$ | 4.58 | 3.61 | 4.41 | 3.87 | 3.62 |
| $L_{\text{dir}}$ | 5.12 | 4.21 | 5.17 | 4.63 | 4.41 |
| $L_{\text{dir,rel}}$ | 0.35 | 0.21 | 0.28 | 0.24 | 0.26 |

Table 2 shows that this learning bias is *not very effective* in guiding the model towards positive semi-definite predictions. Figure 2c shows that the learning bias can have a positive effect during the dynamics of learning, but the final values of $\lambda_\%^-$ are similar whether or not learning bias is used.

**Scaling with dataset size**  Using more training data is effectively an observation bias which should lead to better results for all models. In Figure 3a we plot component loss, $L_{\text{comp}}$, for base CGC model, CGC with data augmentation and positive semi-definite layer (CGC+ve) and MACE model with positive semi-definite layer (MACE+ve). The composition of training datasets is explained in the Appendix. At any dataset size, the equivariant MACE+ve model outperforms the CGC-based models. The CGC model with data augmentation outperforms the base CGC model. The scaling slope was calculated as linear fit on log-log axes and is displayed on the graph. It is evident that the MACE+ve model has the *most favourable scaling*.

## 5.3  CHOICE OF POSITIVE (SEMI-)DEFINITE LAYER AND MACE-SPECIFIC PARAMETERS

We evaluate a number of methods for making the output positive (semi-)definite for MACE model class. [2] The results are displayed in Table 3. We empirically observe that the matrix square method, $A^2$, achieves most favourable results. Moreover, this method also has the lowest associated computational cost. For these reasons, we use the matrix square method throughout the paper whenever "+ve" suffix is used, unless stated otherwise.

**Spherical frequency $L_{max}$ and degeneracy**  One of the most important hyperparameters of the MACE model is the maximum frequency of expansion in spherical basis, $L_{max}$. In Figure 3b we show the sensitivity of model accuracy to the degree of expansion $L_{max}$. Empirically, we observe that degree $L_{max} = 4$ is required to achieve good accuracy. This is in line with the spherical form of the fourth-order stiffness tensor, which contains $L = 4$ components. Moreover, it has been remarked by Joshi et al. (2023) that certain types of highly symmetric graphs require high order of tensors $L$. More specifically, to identify the orientation of neighbourhood with $L$-fold symmetry, at least $L$-order tensors are required. In the Appendix, we show how model which is internally truncated to $L_{max} = 2$ or 3 is unable to predict anisotropic behaviour of a simple cubic lattice (Section A.14, Figure 7).

---

[2]**Positive definite vs semi-definite** The physical principle of energy conservation requires non-negative deformation energy $\psi = \epsilon_{ij} C_{ijkl} \epsilon_{kl}/2 \geq 0 \forall \epsilon$. The case of zero eigenvalue of **C** does not violate energy conservation. A structure whose stiffness tensor has zero eigenvalue is a *mechanism* – it is possible to deform it without exerting any work. However, it is a feature of our dataset that all eigenvalues are positive. Therefore, we have the freedom to try both positive definite and positive semi-definite layers.

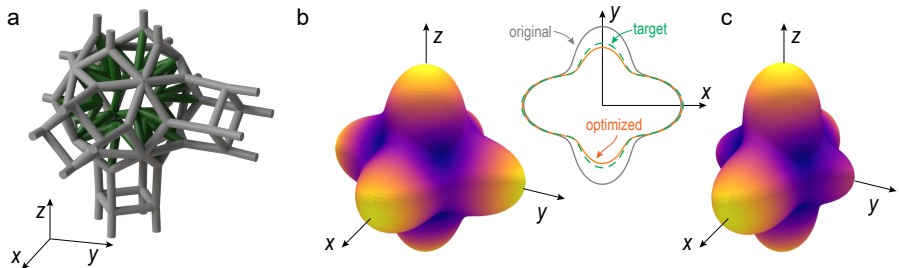

Figure 4: (*a*) Unit cell of the lattice used as the starting point for optimization. (*b*) Original stiffness surface and (*c*) optimized stiffness surface. Inset shows projections into $x - y$ plane of the starting point, optimization target, and the result of ML-based optimization.

**Correlation order $\nu$**  In Figure 3c we show the sensitivity of test error to the body-order of messages. Model with $\nu = 1$ does not contain the equivariant product layer and it equivalent to Tensor Field Network (TFN) with 2-body messages. We observe that increasing the order of messages to three- and four-body ($\nu = 2$ and $3$) significantly reduces error over the test dataset.

## 5.4 Speedup using machine learning methods

Table 4 shows a comparison of inference time for the three classes of investigated GNN models as well as for finite-element calculation. Time is reported for computation of stiffness tensor for 5000 lattices. The tests were run on desktop computer with Intel i7-11700 CPU, 96GB RAM and Nvidia RTX3070 GPU. While the equivariant MACE-based architecture is slower than the more standard CGC- and NNConv-based models, all these models are 3 orders of magnitude faster than FE calculation.

Table 4: Comparison of inference speed for the different ML models and the FE baseline

|  | CGC | NNConv | MACE | FE |
|---|---|---|---|---|
| $t_{5000}(s)$ | 5 | 5 | 15 | $1.1 \times 10^4$ |

## 5.5 Example application: design of an architected material

An important application of architected solids is to achieve complex anisotropic stiffness tensor that cannot be found in existing materials. In Figure 4 we show an example application of our GNN model in a gradient-based optimization scheme for the design of specific stiffness tensor. The starting unit cell, as correctly predicted by the GNN model, has the same stiffness in $x-$ and $y-$directions. The task is to perturb nodal positions to break the $x-y$ symmetry and reduce stiffness in the $y-$direction. We use gradients returned from backpropagation and execute 50 steps of gradient descent algorithm. The optimization scheme produced the desired result with great accuracy as verified post-optimization using FE baseline. Error between the desired output and FE-verified ground truth is $L_{\text{comp}} = 1.84$. Based on the speedup of the GNN model compared to FE, we estimate the GNN-based optimisation to also be 3 orders of magnitude faster than the FE baseline.

## 6 Conclusion

In this work, we present the application of Euclidean equivariant GNNs to the prediction of the $4^{\text{th}}$ order stiffness tensor of architected lattice metamaterials. In addition to the intrinsic equivariance to rigid body rotations and translations, we designed the model to also preserve positive semi-definiteness of the predicted stiffness, in line with energy conservation. We benchmark the model against other architectures that were previously used for property prediction of lattice materials and demonstrate superior performance by all the metrics studied. Finally, we demonstrate a possible downstream use of the model in ML-based structural optimization. Fast and accurate property prediction models, such as the one we are presenting, achieve a significant improvement over the high computational cost of traditional FE methods and they are applicable to tensors beyond the stiffness tensor such as piezo-optical, elasto-optical and the flexoelectric tensors.

REPRODUCIBILITY STATEMENT

To ensure reproducibility and completeness, we include detailed descriptions of the models used, hyperparameters, and data sources in the Appendix. The code is available publicly at github.com/igrega348/energy-equiv-lattice-gnn.git. Datasets are available publicly at doi.org/10.17863/CAM.106854.

ACKNOWLEDGMENTS

The authors acknowledge funding from the UKRI Frontier Research grant "Graph-based Learning and design of Advanced Mechanical Metamaterials" with award number EP/X02394X/1. This work was performed using resources provided by the Cambridge Service for Data Driven Discovery (CSD3) operated by the University of Cambridge Research Computing Service (www.csd3.cam.ac.uk), provided by Dell EMC and Intel using Tier-2 funding from the Engineering and Physical Sciences Research Council (capital grant EP/T022159/1), and DiRAC funding from the Science and Technology Facilities Council (www.dirac.ac.uk). The authors further acknowledge Padmeya Prashant Indurkar for the initial exploration of graph neural networks for lattice metamaterials and are grateful to Angkur Shaikeea for the fruitful discussions throughout the project.

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

# A  APPENDIX

## A.1  SYMMETRIES OF THE STIFFNESS TENSOR

From equation 4, stiffness tensor can be expressed as the derivative of strain energy with respect to strain:

$$\mathbf{C} = C_{ijkl} = \frac{\partial^2 \psi}{\partial \epsilon_{ij} \partial \epsilon_{kl}}$$

The order of $\epsilon_{ij}$ and $\epsilon_{kl}$ is interchangeable, which results in the *major* symmetry for the stiffness tensor: $C_{ijkl} = C_{klij}$.

Furthermore, strain is defined as the *symmetric* gradient of displacement, $\boldsymbol{u}$:

$$\epsilon_{ij} = \frac{1}{2} \left( \frac{\partial u_i}{\partial x_j} + \frac{\partial u_j}{\partial x_i} \right)$$

Therefore, $\epsilon_{ij} = \epsilon_{ji}$, which gives rise to the *minor* symmetry of the stiffness tensor.

All in all, the stiffness tensor $\mathbf{C}$ has both minor and major symmetries:

$$C_{ijkl} = C_{jikl} = C_{ijlk} = C_{klij}$$

As a result, 21 of the $3 \times 3 \times 3 \times 3 = 81$ components of the stiffness tensor are independent.

## A.2  METHODS FOR ENFORCING POSITIVE (SEMI-)DEFINITENESS

Here we outline methods which can be used to enforce positive (semi-)definiteness for $n \times n$ matrices $\mathbb{R}^n \to \mathbb{R}^n$. Section A.4 explains how the $4^{\text{th}}$ order stiffness tensor can be efficiently represented in a matrix form which justifies the use of these methods.

### A.2.1  CHOLESKY-BASED METHOD

Cholesky decomposition is defined for a Hermitian positive-definite matrix $\boldsymbol{A} : \mathbb{R}^n \to \mathbb{R}^n$ as:

$$\boldsymbol{A} = \boldsymbol{L}\boldsymbol{L}^*$$

where $\boldsymbol{L}$ is a lower diagonal matrix and $\boldsymbol{L}^*$ is its conjugate transpose. The diagonal entries of $\boldsymbol{L}$ are positive.

Machine learning methods (Xu et al., 2021; Jekel et al., 2022; Van 't Sant et al., 2023) can use Cholesky factorization as follows. Suppose we require $n \times n$ positive definite matrix $\boldsymbol{A} : \mathbb{R}^n \to \mathbb{R}^n$. A neural network outputs $k = n(n+1)/2$ entries: $a_0, ..., a_k$. They are arranged into lower diagonal matrix L with the diagonal elements passed through a suitable function $\rho : \mathbb{R} \to \mathbb{R}_{>0}$ (such as $\rho(x) = \exp(x)$):

$$\boldsymbol{L} = \begin{bmatrix} \rho(a_0) & 0 & 0 & \dots \\ a_1 & \rho(a_2) & 0 & \dots \\ a_3 & a_4 & \rho(a_5) & \dots \\ \vdots & \vdots & \vdots & \ddots \end{bmatrix}$$

Matrix product $\boldsymbol{L}\boldsymbol{L}^*$ then guarantees a symmetric (Hermitian) positive-definite matrix. Note that if zero eigenvalues are admissible, a different function $\rho : \mathbb{R} \to \mathbb{R}_{\geq 0}$ can be used (e.g. relu).

Such construction, while simple, will not produce equivariant output because components of the matrix $a_0, ...a_k$ are treated independent scalars.

### A.2.2 EIGENVALUE-BASED METHOD

Symmetric matrix $\boldsymbol{A}$ is positive definite iff all its eigenvalues are positive. The eigenvalue-based methods operate on this premise (Jekel et al., 2022).

Suppose we require $n \times n$ positive definite matrix $\boldsymbol{A} : \mathbb{R}^n \to \mathbb{R}^n$. A neural network again outputs $k = n(n+1)/2$ entries: $a_0, ..., a_k$. They are arranged into a symmetric matrix $\boldsymbol{M}$:

$$\boldsymbol{M} = \begin{bmatrix} a_0 & a_1 & a_3 & \dots \\ a_1 & a_2 & a_4 & \dots \\ a_3 & a_4 & a_5 & \dots \\ \vdots & \vdots & \vdots & \ddots \end{bmatrix}$$

Eigenvalue decomposition is performed on this matrix: $\boldsymbol{M} = \boldsymbol{U}\boldsymbol{\Lambda}\boldsymbol{U}^T$. Next, a suitable function $\rho : \mathbb{R} \to \mathbb{R}_{>0}$ is applied to the eigenvalue matrix $\boldsymbol{\Lambda}$:

$$\boldsymbol{\Lambda}^+ = \begin{bmatrix} \rho(\lambda_1) & 0 & \dots \\ 0 & \rho(\lambda_2) & \dots \\ \vdots & \vdots & \ddots \end{bmatrix}$$

and positive definite matrix $\boldsymbol{A}$ is assembled as $\boldsymbol{A} = \boldsymbol{U}\boldsymbol{\Lambda}^+\boldsymbol{U}^T$. Similarly, for positive semi-definiteness, function $\rho : \mathbb{R} \to \mathbb{R}_{\geq 0}$ should be used.

The advantage of this method, as opposed to the Cholesky-based method, is that the geometric representation of eigenvectors is maintained – in other words, if the overall model had been equivariant with respect to vectors in $\boldsymbol{U}$, it will remain equivariant after eigenvalues are made positive. The significant disadvantage of this method is that eigenvalue decomposition is not a stable operation with respect to gradients, which is also noted in the official PyTorch documentation.

### A.2.3 MATRIX POWER AND MATRIX EXPONENTIAL

To avoid the computational complexity and gradient instability of eigenvalue decomposition, we can look for methods which will provide the same result – matrix with positive eigenvalues – without explicitly computing the eigenvalue decomposition. We have experimented with a number of methods which are based on taking even powers of matrix and calculating matrix *exponential*.

**Matrix exponential**   The action of *matrix exponential* on square symmetric $n \times n$ matrix $\boldsymbol{M}$ is

$$\boldsymbol{A} = \text{matrix\_exp}(\boldsymbol{M}) = \boldsymbol{U}e^{\boldsymbol{\Lambda}}\boldsymbol{U}^T$$

i.e. eigenvalues of $\boldsymbol{A}$ are exponentiated eigenvalues of $\boldsymbol{M}$.

The method is usually implemented as an iterative algorithm in which the explicit calculation of eigenvectors is not required. While it is stable with respect to gradients, its execution takes 1.5 times longer than computing eigenvalue decomposition (comparing PyTorch `linalg.matrix_exp(`$\boldsymbol{M}$`)` and `linalg.eigh(`$\boldsymbol{M}$`)`). The key difference between matrix exponential and matrix powers is that it produces strictly positive eigenvalues (and hence positive definite matrix).

**Matrix power**   Even powers of a symmetric $n \times n$ matrix ensure non-negative eigenvalues:

$$\boldsymbol{A} = \boldsymbol{M}^n = \boldsymbol{U}\boldsymbol{\Lambda}^n\boldsymbol{U}^T$$

Therefore, it has the same effect as carrying out the eigenvalue decomposition and raising the eigenvalues to power $n$. However, it has a lower complexity and could be up to 80 times faster (comparing PyTorch `linalg.matrix_power(`$\boldsymbol{M}, 2$`)` and `linalg.eigh(`$\boldsymbol{M}$`)`).

We evaluate the performance of $2^{\text{nd}}$ and $4^{\text{th}}$ power in Section 5.3.

**Truncated matrix exponential**   One of the ways to write matrix exponential is

$$e^{\boldsymbol{M}} = \lim_{k \to \infty} \left( \boldsymbol{I} + \frac{\boldsymbol{A}}{k} \right)^k$$

We evaluate the performance of a positive semi-definite layer for $k = 2$ and 4 in Section 5.3.

An important advantage of these methods as opposed to Cholesky-based methods is that they can be made equivariant. However, that is predicated on using the Mandel notation as opposed to the more traditional Voigt notation, as discussed in the following section.

## A.3 SPECTRUM OF THE 4$^{\text{TH}}$ ORDER TENSOR

As outlined by Lord Kelvin (Thomson, 1856), there are 6 principal strains $\epsilon = \boldsymbol{E}^{(i)}$ such that stress is parallel to strain under that deformation:

$$\sigma\left(\epsilon = \boldsymbol{E}^{(i)}\right) = \mathbf{C} : \boldsymbol{E}^{(i)} = \lambda^{(i)} \boldsymbol{E}^{(i)}$$

where $\lambda^{(i)}$ is a scalar *eigenvalue* of the stiffness tensor and $\boldsymbol{E}^{(i)}, i = 1, ..., 6$ are the six 2$^{\text{nd}}$ order *eigentensors*. See also a more recent text by Basser & Pajevic (2007)

The equivalence between tensor notation using $\mathbf{C}$, $\epsilon_{ij}$, $\sigma_{ij}$ and vector/matrix notation using $\boldsymbol{C}^{(M)}$, $\epsilon^{(M)}$, $\sigma^{(M)}$ provides a way to calculate the eigenvalues and eigentensors of the 4$^{\text{th}}$ order tensor $\mathbf{C}$. The eigenvalues $\lambda^{(i)}$ for tensor $\mathbf{C}$ are the eigenvalues of matrix $\boldsymbol{C}^{(M)}$, and the eigentensors $\boldsymbol{E}^{(i)}$ are obtained by rearranging the eigenvectors of $\boldsymbol{C}^{(M)}$. Suppose $\boldsymbol{x} = \left[\epsilon_{11}, \epsilon_{22}, \epsilon_{33}, \sqrt{2}\epsilon_{23}, \sqrt{2}\epsilon_{13}, \sqrt{2}\epsilon_{12}\right]$ is an eigenvector of $\boldsymbol{C}^{(M)}$. The corresponding eigentensor for $\mathbf{C}$ is

$$\boldsymbol{E} = \begin{bmatrix} \epsilon_{11} & \epsilon_{12} & \epsilon_{13} \\ \epsilon_{12} & \epsilon_{22} & \epsilon_{23} \\ \epsilon_{13} & \epsilon_{23} & \epsilon_{33} \end{bmatrix}$$

## A.4 MANDEL/KELVIN NOTATION VS VOIGT NOTATION

Stress $\sigma_{ij}$ and strain $\epsilon_{ij}$ are 2$^{\text{nd}}$ order symmetric tensors:

$$\boldsymbol{\sigma} = \begin{bmatrix} \sigma_{11} & \sigma_{12} & \sigma_{13} \\ \sigma_{12} & \sigma_{22} & \sigma_{23} \\ \sigma_{13} & \sigma_{23} & \sigma_{33} \end{bmatrix} ; \qquad \boldsymbol{\epsilon} = \begin{bmatrix} \epsilon_{11} & \epsilon_{12} & \epsilon_{13} \\ \epsilon_{12} & \epsilon_{22} & \epsilon_{23} \\ \epsilon_{13} & \epsilon_{23} & \epsilon_{33} \end{bmatrix}$$

They have 6 independent components: three *direct* components (indexed by 11,22,33), and three *shear* components (indexed by 12,13,23). They are often arranged in vector form using *Voigt* notation.

$$\boldsymbol{\sigma}^{(V)} = \begin{bmatrix} \sigma_{11} \\ \sigma_{22} \\ \sigma_{33} \\ \sigma_{23} \\ \sigma_{13} \\ \sigma_{12} \end{bmatrix} ; \qquad \boldsymbol{\epsilon}^{(V)} = \begin{bmatrix} \epsilon_{11} \\ \epsilon_{22} \\ \epsilon_{33} \\ 2\epsilon_{23} \\ 2\epsilon_{13} \\ 2\epsilon_{12} \end{bmatrix}$$

The factor of 2 in front of shear components of strain is to preserve the dot product equivalence for strain energy: in 2$^{\text{nd}}$ order notation, strain energy can be written as the contraction of stress and strain:

$$\psi = \frac{1}{2}\boldsymbol{\sigma} : \boldsymbol{\epsilon} = \frac{1}{2}\boldsymbol{\sigma}^{(V)} \cdot \boldsymbol{\epsilon}^{(V)}$$

Following this notation, the 4$^{\text{th}}$ order stiffness tensor can be represented as $6 \times 6$ matrix $\boldsymbol{C}^{(V)}$ such that $\boldsymbol{\sigma}^{(V)} = \boldsymbol{C}^{(V)}\boldsymbol{\epsilon}^{(V)}$:

$$\boldsymbol{C}^{(V)} = \begin{bmatrix} C_{1111} & C_{1122} & C_{1133} & C_{1123} & C_{1113} & C_{1112} \\ C_{2211} & C_{2222} & C_{2233} & C_{2223} & C_{2213} & C_{2212} \\ C_{3311} & C_{3322} & C_{3333} & C_{3323} & C_{3313} & C_{3312} \\ C_{2311} & C_{2322} & C_{2333} & C_{2323} & C_{2313} & C_{2312} \\ C_{1311} & C_{1322} & C_{1333} & C_{1323} & C_{1312} & C_{1312} \\ C_{1211} & C_{1222} & C_{1233} & C_{1223} & C_{1213} & C_{1212} \end{bmatrix}$$

The disadvantage of this approach is that stress and strain are expressed in contravariant and co-variant bases, respectively, which do not coincide (Helnwein, 2001; Mánik, 2021). This makes the Voigt notation unsuitable for our GNN model. In particular, if the equivariant GNN model outputs

$4^{\text{th}}$ order tensor **C** which are arranged into $6 \times 6$ matrix $\boldsymbol{C}^{(V)}$ using the Voigt notation, and we then apply a positive definite layer (e.g. matrix square), the output matrix *loses equivariance property* (as further explained in the following section).

This issue can be resolved using the Mandel/Kelvin notation. In Mandel notation, the second order stress and strain tensors are also written as 6-dimensional vectors, but they take the following form:

$$\boldsymbol{\sigma}^{(M)} = \begin{bmatrix} \sigma_{11} \\ \sigma_{22} \\ \sigma_{33} \\ \sqrt{2}\sigma_{23} \\ \sqrt{2}\sigma_{13} \\ \sqrt{2}\sigma_{12} \end{bmatrix} \qquad \boldsymbol{\epsilon}^{(M)} = \begin{bmatrix} \epsilon_{11} \\ \epsilon_{22} \\ \epsilon_{33} \\ \sqrt{2}\epsilon_{23} \\ \sqrt{2}\epsilon_{13} \\ \sqrt{2}\epsilon_{12} \end{bmatrix}$$

Strain energy can still be expressed as contraction

$$\psi = \frac{1}{2}\boldsymbol{\sigma} : \boldsymbol{\epsilon} = \frac{1}{2}\boldsymbol{\sigma}^{(M)} \cdot \boldsymbol{\epsilon}^{(M)}$$

Moreover, the norm of stress and strain is preserved under this notation

$$||\epsilon|| = \boldsymbol{\epsilon} : \boldsymbol{\epsilon} = \boldsymbol{\epsilon}^{(M)} \cdot \boldsymbol{\epsilon}^{(M)}; \qquad ||\sigma|| = \boldsymbol{\sigma} : \boldsymbol{\sigma} = \boldsymbol{\sigma}^{(M)} \cdot \boldsymbol{\sigma}^{(M)}$$

The corresponding stiffness tensor $\boldsymbol{C}^{(M)}$ can be written such that $\sigma^{(M)} = \boldsymbol{C}^{(M)}\epsilon^{(M)}$:

$$\boldsymbol{C}^{(M)} = \begin{bmatrix} C_{1111} & C_{1122} & C_{1133} & \sqrt{2}C_{1123} & \sqrt{2}C_{1113} & \sqrt{2}C_{1112} \\ C_{2211} & C_{2222} & C_{2233} & \sqrt{2}C_{2223} & \sqrt{2}C_{2213} & \sqrt{2}C_{2212} \\ C_{3311} & C_{3322} & C_{3333} & \sqrt{2}C_{3323} & \sqrt{2}C_{3313} & \sqrt{2}C_{3312} \\ \sqrt{2}C_{2311} & \sqrt{2}C_{2322} & \sqrt{2}C_{2333} & 2C_{2323} & 2C_{2313} & 2C_{2312} \\ \sqrt{2}C_{1311} & \sqrt{2}C_{1322} & \sqrt{2}C_{1333} & 2C_{1323} & 2C_{1312} & 2C_{1312} \\ \sqrt{2}C_{1211} & \sqrt{2}C_{1222} & \sqrt{2}C_{1233} & 2C_{1223} & 2C_{1213} & 2C_{1212} \end{bmatrix}$$

Using this notation, both stress and strain are expressed in the same orthonormal basis. Contrary to using Voigt notation, we can use the Mandel notation in an equivariant framework. If equivariant GNN model outputs $4^{\text{th}}$ order tensor **C**, we can arrange the components into $6 \times 6$ stiffness matrix, and apply a positive definite layer to this matrix. Importantly, this pipeline *will satisfy equivariance* as shown below.

## A.5  PROOF OF EQUIVARIANCE OF PSD LAYER IN MANDEL NOTATION

Let $\boldsymbol{M}$ be the output (arranged in Mandel notation) of equivariant MACE backbone $\mathcal{M}$ for lattice $\mathcal{L}$

$$\mathcal{M}(\mathcal{L}) = \boldsymbol{M}$$

and let $Q(.\,;\boldsymbol{R})$ represent the rotation given by the rotation matrix $\boldsymbol{R}$ applied on the object, which is the first argument of the function. We postulate that the representation of the rotation group in Mandel basis can be written in terms of matrix $\boldsymbol{R}^{(M)}$ such that the rotated output $\hat{\boldsymbol{M}}$ is given by

$$\hat{\boldsymbol{M}} = \mathcal{M}(Q(\mathcal{L});\boldsymbol{R})) = \boldsymbol{R}^{(M)}\boldsymbol{M}\boldsymbol{R}^{(M),T}$$

After pass through the PSD layer (for instance using matrix square), the output and its rotated version are given by

$$\boldsymbol{A} = \boldsymbol{M}^2 \tag{5}$$

$$\hat{\boldsymbol{A}} = \hat{\boldsymbol{M}}^2 = \boldsymbol{R}^{(M)}\boldsymbol{M}\boldsymbol{R}^{(M),T}\boldsymbol{R}^{(M)}\boldsymbol{M}\boldsymbol{R}^{(M),T} \tag{6}$$

Therefore, the PSD layer constructed in this way is equivariant iff matrix $\boldsymbol{R}^{(M)}$ is orthonormal: $\boldsymbol{R}^{(M),T}\boldsymbol{R}^{(M)} = I$. In this section, we prove both the postulate that the rotation of stiffness tensor in Mandel basis can be expressed using matrix $\boldsymbol{R}^{(M)}$ and the statement that this matrix is orthonormal.

We first consider the basis of representation of stress in Mandel notation. For stress $\sigma^{(M)}$ with components $[u_1, ..., u_6]$, the basis is formed by the following second order tensors

$$\sigma^{(M)} = \begin{bmatrix} u_1 \\ u_2 \\ u_3 \\ u_4 \\ u_5 \\ u_6 \end{bmatrix} = \begin{matrix} u_1 e^{(1)} \otimes e^{(1)} + \\ + u_2 e^{(2)} \otimes e^{(2)} + \\ + u_3 e^{(3)} \otimes e^{(3)} + \\ + \frac{u_4}{\sqrt{2}} \left( e^{(2)} \otimes e^{(3)} + e^{(3)} \otimes e^{(2)} \right) + \\ + \frac{u_5}{\sqrt{2}} \left( e^{(1)} \otimes e^{(3)} + e^{(3)} \otimes e^{(1)} \right) + \\ + \frac{u_6}{\sqrt{2}} \left( e^{(1)} \otimes e^{(2)} + e^{(2)} \otimes e^{(1)} \right) \end{matrix} \tag{7}$$

We now proceed to derive the representation of SO(3) rotation group in Mandel notation. Without loss of generality, we assume that the basis vectors $e^{(1)}, e^{(2)}, e^{(3)}$ are originally aligned with the Cartesian axes. Therefore, the $i$-th component of vector $e^{(j)}$ is equivalent to Kronecker delta: $e_i^{(j)} = \delta_{ij}$. The effect of rotation on the basis vectors $e^{(1)}, e^{(2)}, e^{(3)}$ can be expressed by matrix multiplication with the conventional rotation matrix $R_{ij}$ as

$$\hat{e}_i^{(p)} = R_{ij} e_j^{(p)}$$

where $\hat{e}^{(p)}$ is the basis vector $e^{(p)}$ expressed in the rotated basis and we use standard Einstein summation convention for repeated indices. The elements of the rotation matrix are therefore

$$R_{ij} = \hat{e}^{(i)} \cdot e^{(j)}$$

We now express stress $\sigma$ in the rotated frame $\hat{\sigma}$ in terms of the components of Mandel representation $u_1, ..., u_6$:

$$\hat{\sigma}_{ij} = R_{ip} R_{jp} \left( u_1 e_p^{(1)} e_q^{(1)} + ... + \frac{u_4}{\sqrt{2}} \left( e_p^{(2)} e_q^{(3)} + e_p^{(3)} e_q^{(2)} \right) + ... \right)$$

$$= R_{ip} R_{jp} \left( u_1 \delta_{1p} \delta_{1q} + ... + \frac{u_4}{\sqrt{2}} \left( \delta_{2p} \delta_{3q} + \delta_{3p} \delta_{2q} \right) + ... \right)$$

This can be written as a matrix-vector product in Mandel representation

$$\hat{\sigma}^{(M)} = \boldsymbol{R}^{(M)} \sigma^{(M)}$$

$$= \begin{bmatrix} R_{11}^2 & R_{12}^2 & R_{13}^2 & \sqrt{2}R_{12}R_{13} & \sqrt{2}R_{11}R_{13} & \sqrt{2}R_{11}R_{12} \\ R_{21}^2 & R_{22}^2 & R_{23}^2 & \sqrt{2}R_{22}R_{23} & \sqrt{2}R_{21}R_{23} & \sqrt{2}R_{21}R_{22} \\ R_{31}^2 & R_{32}^2 & R_{33}^2 & \sqrt{2}R_{32}R_{33} & \sqrt{2}R_{31}R_{33} & \sqrt{2}R_{31}R_{32} \\ \sqrt{2}R_{21}R_{31} & \sqrt{2}R_{22}R_{32} & \sqrt{2}R_{23}R_{33} & R_{22}R_{33}+R_{23}R_{32} & R_{21}R_{33}+R_{23}R_{31} & R_{21}R_{32}+R_{22}R_{31} \\ \sqrt{2}R_{11}R_{31} & \sqrt{2}R_{12}R_{32} & \sqrt{2}R_{13}R_{33} & R_{12}R_{33}+R_{13}R_{32} & R_{11}R_{33}+R_{13}R_{31} & R_{11}R_{32}+R_{12}R_{31} \\ \sqrt{2}R_{11}R_{21} & \sqrt{2}R_{12}R_{22} & \sqrt{2}R_{13}R_{23} & R_{12}R_{23}+R_{13}R_{22} & R_{11}R_{23}+R_{13}R_{21} & R_{11}R_{22}+R_{12}R_{21} \end{bmatrix} \begin{bmatrix} u_1 \\ u_2 \\ u_3 \\ u_4 \\ u_5 \\ u_6 \end{bmatrix}$$

Matrix $\boldsymbol{R}^{(M)}$ is the representation of SO(3) rotation in Mandel notation. We can now proceed to derive the corresponding rotation rule for the stiffness matrix $\boldsymbol{C}^{(M)}$. In the original frame,

$$\sigma^{(M)} = \boldsymbol{C}^{(M)} \epsilon^{(M)} \tag{8}$$

while in the rotated frame:

$$\hat{\sigma}^{(M)} = \hat{\boldsymbol{C}}^{(M)} \hat{\epsilon}^{(M)}$$

$$\boldsymbol{R}^{(M)} \sigma^{(M)} = \hat{\boldsymbol{C}}^{(M)} \boldsymbol{R}^{(M)} \epsilon^{(M)}$$

$$\sigma^{(M)} = \boldsymbol{R}^{(M),-1} \hat{\boldsymbol{C}}^{(M)} \boldsymbol{R}^{(M)} \epsilon^{(M)} \tag{9}$$

comparing equations equation 8 and equation 9, we obtain the rotation rule for stiffness matrix $\boldsymbol{C}^{(M)}$:

$$\hat{\boldsymbol{C}}^{(M)} = \boldsymbol{R}^{(M)} \boldsymbol{C}^{(M)} \boldsymbol{R}^{(M),-1} \tag{10}$$

We now proceed to show that matrix $\boldsymbol{R}^{(M)}$ is orthonormal. This can be done by expanding the product $\boldsymbol{R}^{(M),T}\boldsymbol{R}^{(M)}$ and showing that $R_{pi}^{(M)}R_{pj}^{(M)} = \delta_{ij}$, but we choose an alternative route: by considering the double contraction of stress as dot product in Mandel basis.

From equation equation 7, it is straightforward to show that

$$\sigma_{ij}\sigma_{ij} = \sigma_i^{(M)}\sigma_i^{(M)} \qquad \forall \sigma_{ij}$$

We then consider this contraction in rotated basis:

$$\hat{\sigma}_i^{(M)}\hat{\sigma}_i^{(M)} = R_{ip}^{(M)}\sigma_p^{(M)}R_{iq}^{(M)}\sigma_q^{(M)} = \sigma^{(M),T}\boldsymbol{R}^{(M),T}\boldsymbol{R}^{(M)}\sigma^{(M)}$$

Therefore, to show that matrix $\boldsymbol{R}^{(M)}$ is orthonormal, it suffices to show that $\hat{\sigma}_i^{(M)}\hat{\sigma}_i^{(M)} = \sigma_i^{(M)}\sigma_i^{(M)}$.

$$\hat{\sigma}_i^{(M)}\hat{\sigma}_i^{(M)} = R_{ip}R_{jq}R_{ia}R_{jb}\left(u_1\delta_{1p}\delta_{1q} + ... + \frac{u_4}{\sqrt{2}}\left(\delta_{2p}\delta_{3q} + \delta_{3p}\delta_{2q}\right) + ...\right)\left(u_1\delta_{1a}\delta_{1b} + ... + \frac{u_4}{\sqrt{2}}\left(\delta_{2a}\delta_{3b} + \delta_{3a}\delta_{2b}\right) + ...\right)$$

$$= \delta_{pa}\delta_{qb}\left(u_1\delta_{1p}\delta_{1q} + ... + \frac{u_4}{\sqrt{2}}\left(\delta_{2p}\delta_{3q} + \delta_{3p}\delta_{2q}\right) + ...\right)\left(u_1\delta_{1a}\delta_{1b} + ... + \frac{u_4}{\sqrt{2}}\left(\delta_{2a}\delta_{3b} + \delta_{3a}\delta_{2b}\right) + ...\right)$$

$$= u_1^2 + u_2^2 + u_3^2 + u_4^2 + u_5^2 + u_6^2 = \sigma_i^{(M)}\sigma_i^{(M)}$$

where we used the fact that the conventional $3 \times 3$ rotation matrix $R$ is orthonormal $R_{ip}R_{ia} = \delta_{pa}$. Therefore

$$\boldsymbol{R}^{(M),-1} = \boldsymbol{R}^{(M),T}$$

We can now reformulate the rotation rule for stiffness in Mandel notation from equation equation 10:

$$\hat{\boldsymbol{C}}^{(M)} = \boldsymbol{R}^{(M)}\boldsymbol{C}^{(M)}\boldsymbol{R}^{(M),-1}$$

which validates the postulate of this proof. *This combined with equation equation 6 proves that our PSD layer is equivariant.*

### A.6 PROOF OF NON-EQUIVARIANCE OF PSD LAYER IN VOIGT NOTATION

Analogous analysis can be performed for the stress/strain in Voigt basis. However, in Voigt notation, the bases for strain and stress are different:

$$\sigma^{(V)} = \begin{bmatrix} u_1 \\ u_2 \\ u_3 \\ u_4 \\ u_5 \\ u_6 \end{bmatrix} = \begin{array}{c} u_1\boldsymbol{e^{(1)}} \otimes \boldsymbol{e^{(1)}}+ \\ +u_2\boldsymbol{e^{(2)}} \otimes \boldsymbol{e^{(2)}}+ \\ +u_3\boldsymbol{e^{(3)}} \otimes \boldsymbol{e^{(3)}}+ \\ +u_4\left(\boldsymbol{e^{(2)}} \otimes \boldsymbol{e^{(3)}} + \boldsymbol{e^{(3)}} \otimes \boldsymbol{e^{(2)}}\right)+ \\ +u_5\left(\boldsymbol{e^{(1)}} \otimes \boldsymbol{e^{(3)}} + \boldsymbol{e^{(3)}} \otimes \boldsymbol{e^{(1)}}\right)+ \\ +u_6\left(\boldsymbol{e^{(1)}} \otimes \boldsymbol{e^{(2)}} + \boldsymbol{e^{(2)}} \otimes \boldsymbol{e^{(1)}}\right) \end{array}$$

$$\epsilon^{(V)} = \begin{bmatrix} u_1 \\ u_2 \\ u_3 \\ u_4 \\ u_5 \\ u_6 \end{bmatrix} = \begin{array}{c} u_1\boldsymbol{e^{(1)}} \otimes \boldsymbol{e^{(1)}}+ \\ +u_2\boldsymbol{e^{(2)}} \otimes \boldsymbol{e^{(2)}}+ \\ +u_3\boldsymbol{e^{(3)}} \otimes \boldsymbol{e^{(3)}}+ \\ +\frac{u_4}{2}\left(\boldsymbol{e^{(2)}} \otimes \boldsymbol{e^{(3)}} + \boldsymbol{e^{(3)}} \otimes \boldsymbol{e^{(2)}}\right)+ \\ +\frac{u_5}{2}\left(\boldsymbol{e^{(1)}} \otimes \boldsymbol{e^{(3)}} + \boldsymbol{e^{(3)}} \otimes \boldsymbol{e^{(1)}}\right)+ \\ +\frac{u_6}{2}\left(\boldsymbol{e^{(1)}} \otimes \boldsymbol{e^{(2)}} + \boldsymbol{e^{(2)}} \otimes \boldsymbol{e^{(1)}}\right) \end{array}$$

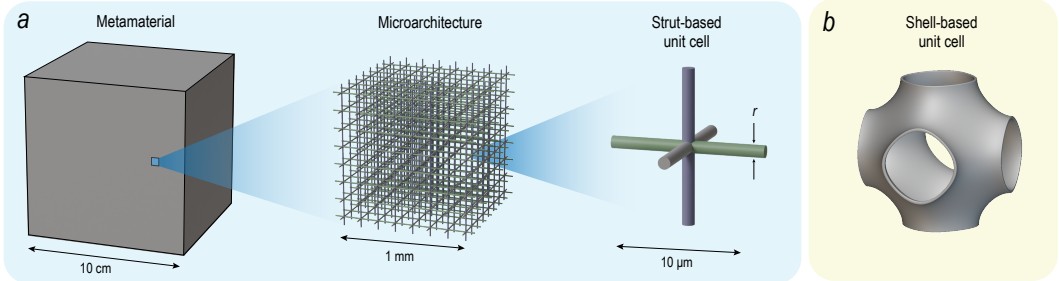

Figure 5: (a) The assembly of millions of unit cells effectively behaves as continuum material, hence the name *meta*material. In this work we assume all unit cells have cylindrical struts with radius $r$. (b) Different types of unit cells could be used, such as triply periodic minimal surfaces (TPMS) which lead to *shell-based* lattices.

It can be shown through analogous process that the corresponding representations of the rotation group are matrices $\boldsymbol{R}^{(V,\sigma)}$ and $\boldsymbol{R}^{(V,\epsilon)}$:

$$\hat{\sigma}^{(V)} = \boldsymbol{R}^{(V,\sigma)}\sigma^{(V)}$$

$$= \left[\begin{array}{ccc|ccc} R_{11}^2 & R_{12}^2 & R_{13}^2 & 2R_{12}R_{13} & 2R_{11}R_{13} & 2R_{11}R_{12} \\ R_{21}^2 & R_{22}^2 & R_{23}^2 & 2R_{22}R_{23} & 2R_{21}R_{23} & 2R_{21}R_{22} \\ R_{31}^2 & R_{32}^2 & R_{33}^2 & 2R_{32}R_{33} & 2R_{31}R_{33} & 2R_{31}R_{32} \\ \hline R_{21}R_{31} & R_{22}R_{32} & R_{23}R_{33} & R_{22}R_{33}+R_{23}R_{32} & R_{21}R_{33}+R_{23}R_{31} & R_{21}R_{32}+R_{22}R_{31} \\ R_{11}R_{31} & R_{12}R_{32} & R_{13}R_{33} & R_{12}R_{33}+R_{13}R_{32} & R_{11}R_{33}+R_{13}R_{31} & R_{11}R_{32}+R_{12}R_{31} \\ R_{11}R_{21} & R_{12}R_{22} & R_{13}R_{23} & R_{12}R_{23}+R_{13}R_{22} & R_{11}R_{23}+R_{13}R_{21} & R_{11}R_{22}+R_{12}R_{21} \end{array}\right]\begin{bmatrix} u_1 \\ u_2 \\ u_3 \\ u_4 \\ u_5 \\ u_6 \end{bmatrix}$$

$$\hat{\epsilon}^{(V)} = \boldsymbol{R}^{(V,\epsilon)}\epsilon^{(V)}$$

$$= \left[\begin{array}{ccc|ccc} R_{11}^2 & R_{12}^2 & R_{13}^2 & R_{12}R_{13} & R_{11}R_{13} & R_{11}R_{12} \\ R_{21}^2 & R_{22}^2 & R_{23}^2 & R_{22}R_{23} & R_{21}R_{23} & R_{21}R_{22} \\ R_{31}^2 & R_{32}^2 & R_{33}^2 & R_{32}R_{33} & R_{31}R_{33} & R_{31}R_{32} \\ \hline 2R_{21}R_{31} & 2R_{22}R_{32} & 2R_{23}R_{33} & R_{22}R_{33}+R_{23}R_{32} & R_{21}R_{33}+R_{23}R_{31} & R_{21}R_{32}+R_{22}R_{31} \\ 2R_{11}R_{31} & 2R_{12}R_{32} & 2R_{13}R_{33} & R_{12}R_{33}+R_{13}R_{32} & R_{11}R_{33}+R_{13}R_{31} & R_{11}R_{32}+R_{12}R_{31} \\ 2R_{11}R_{21} & 2R_{12}R_{22} & 2R_{13}R_{23} & R_{12}R_{23}+R_{13}R_{22} & R_{11}R_{23}+R_{13}R_{21} & R_{11}R_{22}+R_{12}R_{21} \end{array}\right]\begin{bmatrix} u_1 \\ u_2 \\ u_3 \\ u_4 \\ u_5 \\ u_6 \end{bmatrix}$$

and it can be shown that

$$\boldsymbol{R}^{(V,\sigma),-1} = \boldsymbol{R}^{(V,\epsilon),T}.$$

We can now derive the rotation rule for stiffness in Voigt notation as

$$\sigma^{(V)} = \boldsymbol{C}^{(V)}\epsilon^{(V)}$$

$$\hat{\sigma}^{(V)} = \hat{\boldsymbol{C}}^{(V)}\hat{\epsilon}^{(V)}$$

$$\boldsymbol{R}^{(V,\sigma)}\sigma^{(M)} = \hat{\boldsymbol{C}}^{(V)}\boldsymbol{R}^{(V,\epsilon)}\epsilon^{(V)}$$

$$\sigma^{(V)} = \boldsymbol{R}^{(V,\sigma),-1}\hat{\boldsymbol{C}}^{(V)}\boldsymbol{R}^{(V,\epsilon)}\epsilon^{(V)}$$

$$\hat{\boldsymbol{C}}^{(V)} = \boldsymbol{R}^{(V,\sigma)}\boldsymbol{C}^{(V)}\boldsymbol{R}^{(V,\epsilon),-1} = \boldsymbol{R}^{(V,\sigma)}\boldsymbol{C}^{(V)}\boldsymbol{R}^{(V,\sigma),T}.$$

Importantly, matrices $\boldsymbol{R}^{(V,\sigma)}$ and $\boldsymbol{R}^{(V,\epsilon)}$ are not orthonormal $\left(\boldsymbol{R}^{(V,\sigma),T}\boldsymbol{R}^{(V,\sigma)} \neq \boldsymbol{I}\right)$ which implies that using Voigt representation in PSD layer breaks equivariance.

### A.7 LATTICE METAMATERIALS, GRAPH REPRESENTATION OF LATTICE UNIT CELLS AND FE

Figure 5 outlines the concept of *metamaterials*. We consider periodic lattices which are constructed by repeating a predefined building block in three dimensions. This building block is called *unit cell*. When the unit cell is repeated over a distance much longer than its size, there will be millions of unit cells in the sample of interest and its overall response can be characterized by effective material properties.

In this work, we model *strut-based* lattices. There is a clean analogy between such lattice and the mathematical concept of a *graph*. The struts in a lattice can be thought of as *edges*, while their

intersections are *nodes*. We wish to model large samples of metamaterials, which comprise millions of unit cells. Rather than converting such sample into a graph with billions of nodes and edges, we use the concept of *periodicity*: the lattice can be fully defined by its unit cell and we wish to predict the stiffness of the material from the geometry of the unit cell.

All unit cells can be represented in a *reduced* coordinate system, where the unit cell is a unit cube ($0 \leq x_i \leq 1$). The real positions of nodes, *transformed* coordinates $\bar{x}$, can be expressed as an affine transformation of the reduced coordinates: $\bar{x} = Ax$, where $A$ is a suitable matrix. Unit cells that we study in this work range from very simple geometries with just a few nodes to very complex unit cells with hundreds of nodes. We define four node types based on the following conditions:

1. inner nodes, where $0 < x_i < 1 \, \forall i$,

2. face nodes, where $x_i \in \{0, 1\}$ for only one index $i$,

3. edge nodes, where $x_i \in \{0, 1\}$ for two indices $i$,

4. corner nodes, where $x_i \in \{0, 1\}$ for all three indices $i$.

Figure 6 illustrates the 4 node types. Further note that face, edge and corner nodes are shared by 2, 4 and 8 neighboring unit cells, respectively.

The unit cells are representation of the infinite periodic lattice. Therefore, it is possible to shift the window of observation to perceive a different unit cell of the same lattice. This is illustrated in Figure 6a-c. The simple cubic lattice is typically represented as a square with four edges on the boundaries and four *edge* nodes.[3] Displacing the unit cell window, we obtain a view with 4 *face* nodes and 1 *inner node*.

The *fundamental* representation of a lattice is such where only the inner nodes are kept and edges are connected across periodic boundaries. In Figure 6d, we show the fundamental representation of the simple cubic lattice. The lattice has 1 inner node (N0) and 2 fundamental edges (E0, E1). The edges are defined by edge adjacency, and *edge shifts*:

$$\begin{array}{ccc} & \text{adjacency} & \text{shift} \\ E0: & N0 \rightarrow N0; & [1, 0] \\ E1: & N0 \rightarrow N0; & [0, 1] \end{array}$$

If edge has adjacency $N_i \rightarrow N_j$ and shift $t^{(ij)}$, then the edge vector will be $v^{(ij)} = x^{(j)} - x^{(i)} + t^{(ij)}$

Graph representation is obtained when graph is connected according to the *fundamental* edge adjacency and edge shifts are stored with the graph (Fig. 6e).

Note that finite element simulations are run on the windowed representation of lattices, because it makes the handling of periodicity much simpler. In particular, when macroscopic strain $\epsilon_{ij}$ is applied to the material, the following equations are prescribed in FE setup:

$$u_i^B - u_i^A = \sum_j \epsilon_{ij} \left( x_j^B - x_j^A \right)$$
$$\theta_i^B - \theta_i^A = 0$$

## A.8 DATASET

The full dataset contains 8954 base lattices. We selected from this dataset: (i) 7000 training base names, and (ii) 1296 validation/test base names.

Training sets of various sizes are formed as follows:

---

[3]note that intuitively, we might call these nodes *corner* nodes, but to adhere to definitions above, in 2d $x_i \in \{0, 1\}$ for two indices $i$

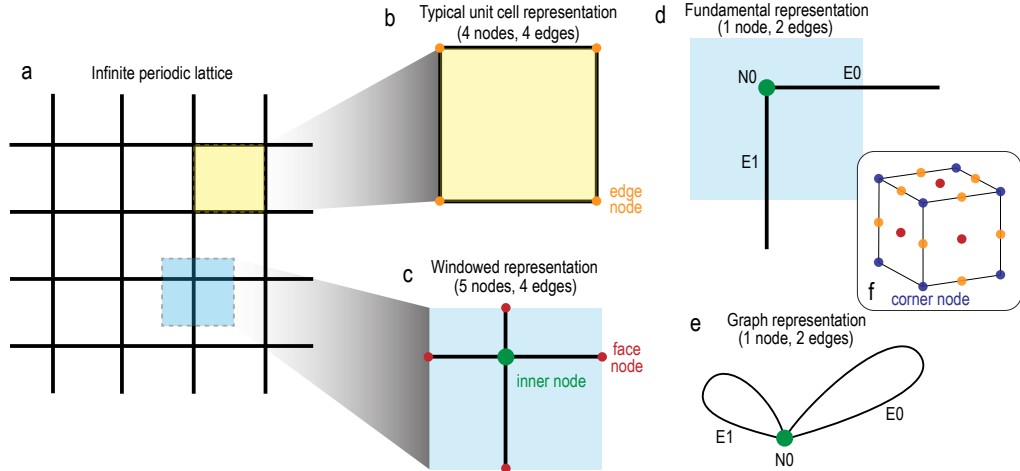

Figure 6: The representation of lattice unit cell in our framework. For simplicity, we show a two-dimensional simple cubic lattice as an example (*a*). The typical representation of the unit cell is a square (*b*). However, to simplify the periodicity conditions in FE calculations, we transform this unit cell to a *windowed* representation (*c*). The fundamental representation connects edges across periodic boundaries and only *inner* node types remain (*d*). The graph used for GNN comes directly from the fundamental representation (*e*). For completeness, we illustrate the remaining *edge* node type in part (*f*).

|  | # graphs | Description |
|---|---|---|
| 0imp_quarter | 1750 | 1750 lattices with no perturbations |
| 0imp_half | 3500 | 3500 lattices with no perturbations |
| 0imp | 7000 | 7000 lattices with no perturbations |
| 1imp | 27847 | 7000 lattices with 1 realization at 0.0,0.02,0.04,0.07 levels |
| 2imp | 48681 | 7000 lattices with 1 realization at 0.0 and 2 realizations at 0.02,0.04,0.07 levels |
| 4imp | 90336 | 7000 lattices with 1 realization at 0.0 and 4 realizations at 0.02,0.04,0.07 levels |

Note that three relative densities (strut radii) of each distinct geometry are used.

## A.9 GRAPH ATTRIBUTES

**Vanilla CGC** In the base CGC model, we use the same node, edge and graph features as Meyer et al. (2022) with the addition of strut radius. Node features of the graph are nodal positions:

$$\boldsymbol{h} = [x_1, x_2, x_3]$$

Edge features are unit vector, length, and radius:

$$\boldsymbol{e} = [u_1, u_2, u_3, L, r]$$

**Augmented GCG-based models** We wish to have a model which is invariant to rigid body translation, therefore we drop nodal positions from node features. The following input features are used.

$$\boldsymbol{h} = [1]$$
$$\boldsymbol{e} = [u_1, u_2, u_3, L, r]$$

## A.10 ARCHITECTURE

We consider lattices as geometric graphs, with node positions, $\boldsymbol{x}_i \in \mathbb{R}^3$, edge adjacency, $\{\{i, j\}\}$, edge shifts[4], $\boldsymbol{u}_i \in \mathbb{R}^3$, and edge thickness, $r_{ij} \in \mathbb{R}^+$. Note that edge adjacency is a multiset as there can be multiple edges between the same set of nodes. The role of edge shifts is to account for periodic connections. Further detail can be found in the Appendix. The model we develop acts in three steps, first the embedding, then $S$ layers of MACE, and finally the readout.

---

[4]As defined in Section A.7 of the Appendix

**Embeddings**   The length and thickness of the edges are encoded using Gaussian embeddings with 6 bases: $G_L, G_r : \mathbb{R} \rightarrow \mathbb{R}^6$. They are then concatenated and used as edge attributes

$$z_{ij} = \left\{ e^{-\gamma_c(\|x_i - x_j\|_2 - \mu_c)^2} \right\}_c \oplus \left\{ e^{-\gamma_c(r_{ij} - \mu_c)^2} \right\}_c \tag{11}$$

where $\oplus$ denotes concatenation and $(\gamma_c, \mu_c)$ are a collection of fixed parameters of the Gaussians and they depend on the number of bases. The edge vectors are expanded in a spherical basis up to $L_{max} = 4$. Unlike atoms of various elements in chemistry, all our nodes are of the same type. Therefore, the node features are initialized as ones and are expanded to the desired number of dimensions using a linear layer: $h_i^{(0)} = w_i$.

**MACE layer**   At each layer $s$ of MACE, edge embedding $\phi_{ij}^{(s)}$ are formed by taking the tensor product between the node features $h_j$ and the edge vectors expanded in a spherical basis. This tensor product is weighted with a non-linear function of the edge attributes $z_{ij}$,

$$\phi_{ij,k\eta_1 l_3 m_3}^{(s)} = \sum_{l_1 l_2 m_1 m_2} C_{\eta_1, l_1 m_1 l_2 m_2}^{l_3 m_3} R_{k\eta_1 l_1 l_2 l_3}^{(s)}(z_{ij}) Y_{l_1}^{m_1}(\hat{x}_{ij}) h_{j,kl_2 m_2}^{(s)} \tag{12}$$

Where $k$ indexes feature channels, $l, m$ index angular momenta, $C_{l_3 m_3 \eta_1, l_1 m_1 l_2 m_2}$ are the Clebsch-Gordan coefficients that enforce the equivariance[5], and $\eta$ indexes combinations of $lm$ which preserve equivariance. The Atomic Basis $A_i^{(s)}$ of the node $i$ at layer $s$ is constructed by summing edge embeddings over the edges of $i$:

$$A_{i,kl_3 m_3}^{(s)} = \sum_{\tilde{k}, \eta_1} W_{k\tilde{k}\eta_1 l_3}^{(s)} \sum_{j \in \mathcal{N}(i)} \phi_{ij, \tilde{k}\eta_1 l_3 m_3}^{(s)} \tag{13}$$

A tensor product is applied to the Atomic Basis $\nu$ times to increase the body order of the feature, and the resulting features are symmetrized using a set generalized Clebsch-Gordan coefficients $\mathcal{C}_{\eta_\nu lm}^{LM}$.

$$\boldsymbol{B}_{i,\eta_\nu kLM}^{(s),\nu} = \sum_{lm} \mathcal{C}_{\eta_\nu lm}^{LM} \prod_{\xi=1}^{\nu} A_{i,kl_\xi m_\xi}^{(s)} \tag{14}$$

where $\boldsymbol{B}_{i,\eta_\nu kLM}^{(s),\nu}$ are called *sketched* product basis. The $\boldsymbol{B}-$features are then linearly mixed to form a many-body message,

$$m_{i,kLM}^{(s)} = \sum_{\nu} \sum_{\eta_\nu} W_{z_i \eta_\nu kL}^{(t),\nu} \boldsymbol{B}_{i,\eta_\nu kLM}^{(s),\nu} \tag{15}$$

Finally the message is used to update the next node features using an update function.

**Readout**   After $S$ layers of MACE, we use an equivariant non-linear readout followed by global graph pooling. Invariance to tessellation is maintained by *mean* pooling operation which ensures that the predicted stiffness will not grow if nodes with identical neighborhoods are added to the graph. [6] Finally, another linear layer outputs two scalars, two $l = 1$ vectors and one $l = 4$ vector, corresponding to the spherical component of a 4th order tensor with the correct permutation symmetry. This is converted to Cartesian basis and assembled into Mandel notation to form the final matrix output $A$ (see the Appendix for details).

**Positive Semi-Definite Layer**   The positive semi-definite layer is the key step to ensure that the final output is positive semi-definite, in line wih the law of energy conservation. We evaluate a number of methods to make the stiffness tensor positive semi-definite. These include taking even powers of the matrix, $\boldsymbol{A}^2$ and $\boldsymbol{A}^4$, matrix exponential, and its truncated versions, $e^{\boldsymbol{A}}, (\boldsymbol{I} + \boldsymbol{A}/2)^2$, $(\boldsymbol{I} + \boldsymbol{A}/4)^4$.

---

[5]For further details, see the original MACE paper by Batatia et al. (2022)

[6]This is another fundamental inductive bias of our model: the outputs for a unit cell and its $n \times n \times n$ tessellation are identical.

## A.11 Training and evaluation details

**Definitions of loss metrics** If $C_p = C_{pij}$ and $\tilde{C}_p = \tilde{C}_{pij}$ are the predicted and target stiffnesses for lattice $p$ (in Mandel notation), respectively, and $\mathbf{C}_p = C_{pijkl}$ and $\tilde{\mathbf{C}}_p = \tilde{C}_{pijkl}$ are the predicted and target stiffnesses for lattice $p$ (in $4^{\text{th}}$ order notation), respectively, the loss used during training is calculated as:

$$\gamma_p = \frac{1}{36}\sum_{ij}\tilde{C}_{pij}\tilde{C}_{pij} \quad L_{\text{comp},p} = \sum_{ij}\left(C_{pij} - \tilde{C}_{pij}\right)^2 \quad L_{\text{train}} = \frac{1}{\mathcal{B}}\sum_p \frac{L_{\text{comp},p}}{\gamma_p} \quad (16)$$

where $\mathcal{B}$ denotes the total number of lattices $p$, $\gamma_p$ the mean stiffness, $L_{\text{comp},p}$ the component loss and $L_{\text{train}}$ the overall loss. For testing purposes, in addition to loss $L_{\text{comp}}$, we define directional loss, $L_{\text{dir}}$, and relative directional loss, $L_{\text{dir,rel},p}$ which are calculated using $N = 250$ random directions on the unit sphere ($\boldsymbol{d}_q, q = 1...N$):

$$L_{\text{dir},p} = \frac{1}{N}\sum_q \left| \sum_{ijkl}\left(C_{pijkl} - \tilde{C}_{pijkl}\right)d_{qi}d_{qj}d_{qk}d_{ql} \right| \qquad L_{\text{dir,rel},p} = L_{\text{dir},p}/\sqrt{\gamma_p} \quad (17)$$

with $L_{\text{dir},p}$ the directional loss and $L_{\text{dir},p}$ the relative directional loss. We further report the percentage of lattices with negative eigenvalues, $\lambda_\%^-$, and equivariance loss, $L_{\text{equiv}}$ which is calculated as follows. We choose $S$ random orientations (here $S = 10$) parameterized by corresponding rotation matrices $\boldsymbol{R}^{(s)} = R_{ij}^{(s)}$, ($s = 1...S$). The predicted stiffness tensor in the original orientation is $\hat{\mathbf{C}}^{(p)}$. For each lattice in the test dataset $\mathcal{L}^{(p)}$, we calculate the predictions for each of the 10 rotations: $C_{ijkl}^{(p,s)}$. Equivariance loss is defined as

$$L_{\text{equiv}} = \frac{1}{SN\mathcal{B}}\sum_{pqs}\left| \sum_{ijkl}\left(Q\left(\hat{\mathbf{C}}^{(p)}; \boldsymbol{R}^{(s)}\right)_{ijkl} - C_{ijkl}^{(p,s)}\right)d_{qi}d_{qj}d_{qk}d_{ql}\right|$$

Note that the equivariance loss is calculated purely from predictions, disregarding the mismatch from the ground truth.

All models were trained on a single NVIDIA A100 GPU with 80GB of memory. The training routines were handled by Pytorch Lightning. Specifics vary between models and are outlined below.

### A.11.1 CGC and mCGC

Hyperparameters were searched on a grid (Table 5). Every experiment was run with constant learning rate for up to $100\,000$ steps. Optimizer AdamW was used with settings $(\beta_1, \beta_2) = (0.9, 0.999)$, $\epsilon = 1 \times 10^{-8}$, weight decay=$1 \times 10^{-8}$. Validation loss was checked every 100 steps and training was stopped by early stopping callback with patience 50 if validation loss has not reduced.

The setup of the *vanilla* CGC and mCGC followed as closely as possible the methods used by Meyer et al. (2022). Key differences between our training routines for CGC-based models and the vanilla versions are as follows.

- Static 7-fold data augmentation was used for training set. Each lattice was rotated by $90\deg$ around the $x-$, $y-$ and $z-$axes, and mirrored about the $x - y$, $x - z$, and $y - z$ planes. This added six more versions of the lattice into the dataset.

- The 21 components of stiffness tensor were independently normalized to lie between 0 and 1.

- Optimizer RAdam was used

- Loss `smooth_l1` was used

### A.12 NNConv

In model NNConv, linear learnable layers were used as node and edge feature embeddings to increase latent dimensionality. The message passing NNConv layer was composed of 3-layer neural

Table 5: Hyperparameter search for CGC and mCGC models

| | |
|---|---|
| ndim | 16, 32, 64, 128, 256, 512 |
| message passes | 2, 3, 4, 6 |
| aggregation | min, max, mean |
| batch size | 256 |
| lr | 0.0003, 0.001, 0.003, 0.01 |
| # parameters | 16K – 10M |

Table 6: Hyperparameter search for NNConv models

| | |
|---|---|
| ndim | 16, 32, 64 |
| message passes | 2, 3, 4, 6 |
| aggregation | min, max, mean |
| batch size | 256 |
| lr | 0.0003, 0.001, 0.003, 0.01 |
| # parameters | 22K – 1.6M |

network with ReLU nonlinearity. "Sum" aggregation was used to aggregate messages from neighbors, after which ReLU layer was applied. The layers of message passing were not shared, but independent. A residual connection between layers was used. Hyperparameters were searched on a grid (Table 6). Every experiment was run with constant learning rate for up to $100\,000$ steps. Optimizer AdamW was used with settings $(\beta_1, \beta_2) = (0.9, 0.999)$, $\epsilon = 1 \times 10^{-8}$, weight decay=$1 \times 10^{-8}$. Validation loss was checked every 100 steps and training was stopped by early stopping callback with patience 50 if validation loss has not reduced.

### A.12.1 MACE

Hyperparameters for MACE-based models were searched on a grid in Table 7. Every experiment was run with a constant learning rate for up to $30\,000$ steps. Optimizer AdamW was used with settings $(\beta_1, \beta_2) = (0.9, 0.999)$, $\epsilon = 1 \times 10^{-8}$, weight decay=$1 \times 10^{-8}$. A smaller batch size of 64 was used because of the higher memory requirements of the MACE model. To maintain consistency with the batch size of 256 from CGC models, gradient accumulation over 4 batches was used. Validation loss was checked every 100 steps and training was stopped by early stopping callback with patience 50 if validation loss has not reduced. Value-based gradient clipping was used with cutoff 10.0.

### A.13 PRIMAL, DUAL AND COMBINED GRAPHS

The choice of graph over which message passing is run is important. For instance, some studies in the mechanics community use the dual graph where the centres of lattice cells are converted to nodes, and the neighbouring cells are connected by graph edges.(Karapiperis & Kochmann, 2023)

Meyer et al. (2022) combine the primal graph, with a *line graph*. The original (primal) graph can be converted to a line graph as follows. The nodes of line graph are the edges of primal graph. Two

Table 7: Hyperparameter search for MACE models

| | |
|---|---|
| hidden dim | 8, 16, 32, 64 |
| readout dim | 8, 16, 32 |
| message passes | 2 |
| aggregation | mean |
| batch size | 64 |
| lr | 0.0003, 0.001, 0.003 |
| # parameters | 50K – 600K |

Table 8: Performance of CGC model for *primal*, *dual*, and *combined* graph representation of lattice unit cells

| | Static augmentation | | | Dynamic augmentation | | |
|---|---|---|---|---|---|---|
| | primal CGC | dual | combined mCGC | primal CGC | dual | combined mCGC |
| $L_{\text{comp}}$ | 7.81 | 11.90 | 8.49 | 4.63 | 9.10 | 5.36 |
| $L_{\text{dir}}$ | 8.31 | 15.46 | 8.71 | 5.29 | 12.03 | 5.84 |
| $L_{\text{dir,rel}}$ | 0.38 | 1.14 | 0.42 | 0.25 | 0.76 | 0.27 |
| $\lambda_\%^-$ | 10 | 1 | 2 | 26 | 0 | 28 |

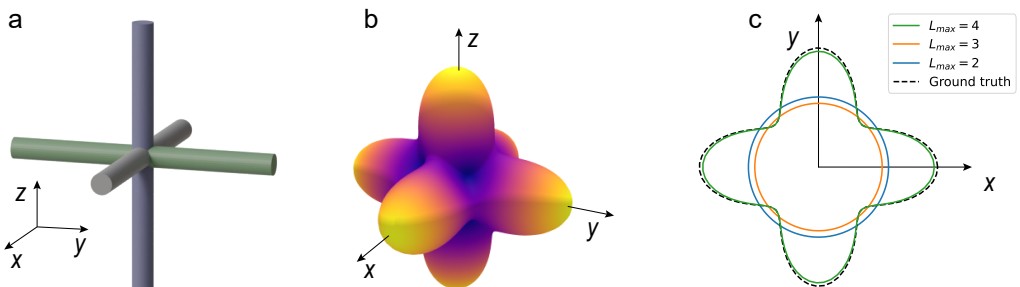

Figure 7: (*a*) Unit cell and (*b*) the true stiffness surface of *simple cubic* lattice. (*c*) Projection into $x - y$ plane and comparison of models with various $L_{max}$.

nodes in the line graph are connected if the two corresponding edges of the primal graph meet at a node.

Here we investigate the performance of GNN based on the choice of graph over which message passing is done. Table 8 shows the results for various models and training strategies. *Primal* and *combined* correspond to models CGCNN and mCGCNN from Meyer et al. (2022). *Dual* is message passing done purely on the line graph. *Static augmentation* corresponds to the training routine from Meyer et al. (2022) as described in Section A.11.1. *Dynamic augmentation* corresponds to our training routine whereby each time a lattice is retrieved from the dataset, it is obtained at a different orientation.

In summary, we empirically do not see any benefit of incorporating the line graph into our model. Therefore, we do not consider these models in the main text.

### A.14 TRAINING WITH $L_{max} < 4$ AND DEGENERACY OF HIGHLY-SYMMETRIC LATTICES

In Figure 7 we show the unit cell of the *simple cubic* lattice. The lattice has a high degree of symmetry which has profound consequences for message passing. If the maximum degree of spherical expansion, $L_{max}$, inside the model is lower than 4, the model is restricted to fitting an isotropic stiffness tensor for this lattice. When the $L_{max} \geq 4$, the anisotropy can be captured.

Note that even when the model is trained with $L_{max} < 4$, it still needs to output a fourth-order tensor whose spherical form includes $L = 4$ component: $2 \times 0e + 2 \times 2e + 1 \times 4e$. [7] We enable this by incorporating a tensor product expansion layer. Suppose the message passing is done up to $L_{max} = 2$. After message passing and graph pooling, each graph has features of the form $N \times 0e + N \times 1o + N \times 2e$, where $N$ is the number of channels ($N$ chosen even). We split the channels into two sets of size $m = N/2$ and do a tensor product between them:

$$[m \times 0e + m \times 1o + m \times 2e] \otimes [m \times 0e + m \times 1o + m \times 2e] \rightarrow [(3m^2) \times 0e + (4m^2) \times 2e + (m^2) \times 4e]$$

The output is passed through a linear layer with learnable weights which reduces it to the correct dimensionality $2 \times 0e + 2 \times 2e + 1 \times 4e$.

---

[7]The notation used here is specific to the software implementation of e3nn (Geiger et al., 2022).

