# OpenReview forum: "Energy-conserving equivariant GNN for elasticity of lattice architected metamaterials"
_ICLR.cc/2024/Conference — ICLR 2024 poster_

### Official Review · Reviewer_uV3y · 2023-10-26

**Soundness:** 3 good
**Presentation:** 3 good
**Contribution:** 2 fair
**Rating:** 6
**Confidence:** 3

**Summary:**

## Summary

The paper presents an extension to the [MACE](https://proceedings.neurips.cc/paper_files/paper/2022/file/4a36c3c51af11ed9f34615b81edb5bbc-Paper-Conference.pdf) model by adding a matrix power layer to ensure the positive semidefinite (PSD) nature of the output tensor. Although the idea of maintaining PSD properties might be of value, the paper's contributions are incremental at best. Most notably, the claim about "high-order rotational equivariance" is not an original contribution but belongs to the original MACE model. The paper severely suffers from a lack of organization and clarity in writing. Hence, I recommend a rejection of this submission.

## Detailed Comments

### 1. Lack of Definitions and Citations

a) The term "edge shifts" is used without a definition or reference.
b) The suffix 'lb' used in the context of training methods is not defined.
c) Clebsch–Gordan coefficients are mentioned without proper citation.
d) On page 17, there are numbers in parentheses next to "Young's modulus," "shear modulus," and "Poisson's ratio," without any explanation.

### 2. Unclear Illustrations and Descriptions

a) The paper claims that energy conservation equates to a PSD tensor, but this is tucked away in a footnote. This claim needs to be explained in the main text.
b) The choice of the optimal 've' method, which is squaring matrix A, is not mentioned until late in the results section. This should be mentioned in the methodology section.
c) Only absolute errors are presented without giving the context of ground truth magnitudes or relative errors. It's unclear for the readers to know if this method is accurate or not at all.
d) It's unclear if Figure 1b is a plot for ground truth or predictions.

### Flaws in Finite Element Method (FEM) Introduction

a) The paper mentions that FEM has "~10^9 elements" but does not state the computation time for such a scale. Furthermore, GNNs cannot handle such scales on a single GPU, making the comparison unfair. Instead, try to report a wall time comparison between your method and FEM for the same dataset.
b) The statement that FEM ensures force equilibrium is incorrect; it is the underlying PDE that ensures this.
c) Similarly, FEM itself does not ensure PSD properties; this is ensured by the constitutive model.
d) The paper incorrectly claims that FEM is rotationally equivariant. Special treatments are needed to achieve rotational equivariance in FEM.

## Conclusion

The paper presents an incremental extension to the MACE model with a focus on preserving the PSD properties of the output tensor. However, the paper lacks clarity in writing and organization, and its contributions are limited. As a result, I recommend a borderline reject for this submission.

The authors can also consider a workshop or journal submission that focuses on this area.

##
After rebuttal, I think the writing quality and the motivation has been more clear. Hence, decided to improve to 6.

**Strengths:**

See above

**Weaknesses:**

See above

**Questions:**

See above

---

> ### Author Response · Authors · 2023-11-18
> **Response to Reviewer uV3y**
>
> We thank the reviewer for their helpful comments.
> It is clear that the reviewer strives to ensure the quality of the submission.
> We made our best effort to address the reviewer's comments in order to improve the quality of the paper.
>
> In response to the reviewer's comments:
>
>
> 1.  The reviewer points out a lack of organization and clarity in writing.
> We have improved the organization of the paper.
> Several sections (MACE details, training details) were moved to the Appendix, and Methods section was expanded to include the details of the PSD layer and Mandel representation.
>
> 2.  With regards to reviewer's comment "1. Lack of Definitions and Citations''. Term edge shifts is defined in section A.5 in the Appendix. Suffix 'lb' is defined in the text (paragraph **Energy conservation learning**). It is mentioned in text that Clebsch-Gordan coefficients  enforce the equivariance of the layer. A formal introduction of Clebsch-Gordan coefficients is beyond the scope of this paper -- we now point the reader to the MACE paper for more detail.
> The numbers by "Young's modulus" etc are now expanded in words.
>
> 3.  With regards to reviewer's comment "2. Unclear Illustrations and Descriptions''. The claim that energy conservation equates to PSD tensor is now explained in main text, as is the equivalence between the positive-definite fourth-order tensor and positive-definite stiffness matrix in Mandel notation.
> The Methods section now mentions the various methods for ensuring positive semi-definiteness.
> It is not quite true that only absolute errors are presented without the context of ground truth magnitudes or relative errors. We also report error $L_{{dir,rel}}$ which is directional error normalized by the square root of the tensor norm.
> We edited the caption of Figure 1 to clarify that the plotted surface corresponds to ground truth.
>
> 4.  With regards to reviewer's comment "Flaws in Finite Element Method (FEM) Introduction''. We have included section **Speedup using machine learning methods** which compares runtime of ML methods that we used and FE.
> The ML methods are 3 orders of magnitude faster than FE baseline.
> The reviewer criticises our statements that (i) FE enforces force equilibrium, and (ii) FE gives PSD output.
> We agree with the reviewer that it is the underlying PDE and constitutive law, respectively, that enforce those principles.
> However, in this context, the *FE solution* for a lattice does indeed satisfy these principles.
> Moreover, no special treatments are required for equivariance.
>
>
> We thank the reviewer for the helpful comments. We made relevant changes to the manuscript which improve its quality.
> We believe that the contributions of this paper are significant on two levels:
> (i) it is not trivial to enforce positive semi-definiteness of a 4-th order tensor while maintaining equivariance (Reviewer azcZ pointed out a need for formal mathematical proofs, which we now include in the manuscript.)
> (ii) The paper is submitted in the area of *applications to physical sciences*, and another key contribution is that we are making the dataset publicly available. There has been a lack of datasets and corresponding method development for higher-order tensors, which we are hoping to address by this work.

---

> > ### Comment · Reviewer_uV3y · 2023-11-20
> > **Reply**
> >
> > I thank the authors for replying, here is more follow-up
> >
> > 1. My concern mentioned in the summary that "the novelty is limited" is not addressed. I noticed you replied to Reviewer TDG7
> > though on this. I think that is a valid point.
> >
> > 2. Your reply 2. mentions that many of the definitions are in the appendix. This is not a good way of expressing, please at least mention their full definition when they 1st appear. You can move further details and proofs in the appendix though for being concise.
> >
> > 3. Your reply 3. equivariance also relies on the constitutive model. Non-linear models without the special treatment on the rotation can cause trouble on this point. However, I admit in your context, this may not be an issue. It's better to be more precise.
> >
> > Based on your reply to other reviewers, I am improving my score to 6. Please refine a bit more according to the 2,3 items above.

---

### Official Review · Reviewer_TDG7 · 2023-10-29

**Soundness:** 3 good
**Presentation:** 2 fair
**Contribution:** 3 good
**Rating:** 5
**Confidence:** 4

**Summary:**

This paper introduces a specially design graph neural network that well tackles the physical constraints in lattice architected metamaterials. To be specific, the model guarantees by design SE(3)-equivariance and energy conservation. The latter is fulfilled by the proposed positive semi-definite layer that overcomes the limits of previously proposed approaches using Cholesky factorization and eigenvalue decomposition. The model is examined on a dataset ``limp'' showcasing that MACE is superior when equipped with the proposed PSD layer.

Update: contribution raised from fair to good, overall score raised from 3 to 5 due to additional experiments on material design.

**Strengths:**

1. The motivation of the paper is clear: designing a GNN that fulfills the inductive biases for lattice architected metamaterials.

2. The proposed approach is clean and easy to follow.

**Weaknesses:**

1. The novelty is somewhat limited in the sense that the proposed model is a direct utilization of MACE, equipped with the proposed layer that satisfies PSD.

2. The dataset is a bit limited and the experiments lack of comparison with other methods. If there are no significant deep learning based (or even GNN-based) methods to compare, it would also help a lot if any FE-based method could be involved. The readers may be curious about how the proposed model can achieve better accuracy/efficiency tradeoff compared with traditional solvers.

3. The implication of the practical usage of the method is limited. Since this paper is developed purely based on practical considerations, it would be better if the paper could show how this model can help in other related/similar tasks or downstream tasks that could potentially benefit from this method., other than just predict the stiffness tensor.

**Questions:**

1. How does the method perform compared with other solvers, deep learning-based or even not?

2. Are there other tasks/datasets that can benefit from the development of such GNN-based method that is specifically designed to regress on the stiffness?

3. Even if the paper discusses the limitations of previous Cholesky/eigenvalue decomposition-based methods that permit PSD constraint, it would help a lot if this point is also verified by experimental results/ablation studies.

---

> ### Author Response · Authors · 2023-11-18
> **Response to Reviewer TDG7**
>
> We would like to thank the reviewer for taking the time to review the paper and providing relevant comments that will help us improve the quality of the submission.
>
> We are addressing all reviewer's comments:
>
>
> 1.  The reviewer states that
>
> > The novelty is limited in the sense that the proposed model is a direct utilization of MACE, equipped with the proposed layer that satisfies PSD.
>
> We recognize that the reviewer is worried about the lack of ML-specific novelty of this work.
> We would like to address these concerns on two levels:
> (i) it is not trivial to enforce positive semi-definiteness of a 4-th order tensor while maintaining equivariance (Reviewer azcZ pointed out a need for formal mathematical proofs, which we now include in the manuscript.) Therefore, we believe that this constitutes significant contribution to the community.
> (ii) The paper is submitted in the area of *applications to physical sciences*, and another key contribution is that we are making the dataset publicly available. There has been a lack of datasets and corresponding method development for higher-order tensors, which we are hoping to address by this work.
>
> 2.  It is true that the dataset lacks comparisons with other methods -- that is because there is presently a lack of methods which focus on lattices and 4-th order tensors.
> The two previously presented GNN methods for lattices are based on CGC and NNConv, which are the models against which we benchmark our MACE-based model.
> Moreover, we have included runtime comparisons for the three model families, and for FE baseline.
> We show that the ML models achieve a speedup of 3 orders of magnitude.
>
> 3.  We thank the reviewer for pointing out other tasks that could benefit from this methods other than just the prediction of the stiffness tensor.
> The presented methods is applicable to any 4-th order tensor. Examples include diffusion tensor in MRI,  piezo-optical tensor and the elasto-optical tensor.
> We have incorporated this statement in the Abstract and Conclusion.
>
> 4.  We do not provide benchmarks with Cholesky/eigenvalue based methods because the Choleksy method fails on the basic requirement of equivariance, and the eigenvalue-based method is unstable during training due to the underlying instability of gradients of eigenvector computation.

---

> > ### Comment · Reviewer_TDG7 · 2023-11-21
> > **Response**
> >
> > Thank the authors for the response.
> >
> > I am still concerned on the novelty of the paper, especially considering potential limited audience at a venue like ICLR--I think the paper would benefit much more if the method or the target problem would have implications to benefit a wider range of tasks, or at least demonstrate any downstream applications besides simply measuring the prediction error of the stiffness tensor, e.g., how the method would help in structural design/optimization.
> >
> > Since it is a first work that brings the task to ML venue, I have to place a little bit higher standard on the scope of the empirical evaluation. I think a more comprehensive benchmark protocol will help more in advancing ML for material science community.

---

> > > ### Author Response · Authors · 2023-11-21
> > > **Response to Reviewer TDG7 2.0**
> > >
> > > We thank the reviewer for highlighting their outstanding reservations.
> > > The reviewer mentions that the paper would benefit from a demonstration of a downstream application, e.g. how the method would help in structural design/optimization.
> > >
> > > Based on this feedback, we revised the manuscript and we included the section **Example application: design of an architected material** -- it is an important application of architected solids to achieve complex anisotropic stiffness tensors that cannot be found in existing materials.
> > > We now use our GNN model demonstrate this downstream application.

---

> > > > ### Comment · Reviewer_TDG7 · 2023-11-22
> > > > **Further response**
> > > >
> > > > I appreciate the additional experiment by the authors on material design. I thank the authors for the efforts. I believe the paper benefits a lot from such interesting illustrations and is able to attract broader audience and inspires more future works.
> > > >
> > > > For now I am raising the score to 5 but is open to further increasing the score in reviewer discussions barring any other concerns raised during the process.

---

### Official Review · Reviewer_kbo4 · 2023-10-31

**Soundness:** 3 good
**Presentation:** 3 good
**Contribution:** 3 good
**Rating:** 8
**Confidence:** 4

**Summary:**

This paper introduces higher-order energy-conserving SE(3) equivariant GNNs which build upon the MACE architecture. These GNNs are applied to lattices, i.e. architected metamaterials. The new features of the model are conservation of energy and SE(3) equivariant predictions of the a 4th order stiffness tensor.  For the stiffness tensor positive-definiteness is ensured. Experiments compare against existing deep learning models of Crystal Graph Convolution (CGC) and NNConv.

**Strengths:**

- This paper introduces a novel and very interesting application of SE(3) equivariant GNNs.
- Positive semi-definiteness of stiffness tensor is a interesting new tool for physics-based machine learning.
- The paper contains all important information to follow (general physics background, no material-design expert).
- The evaluation scheme is quite solid and the different training methods are sound.

**Weaknesses:**

- It seems that most of the important information which is novel to the deep learning community is put into the appendix. For example, definitions of stress and strain tensors, or their relation to the stiffness tensor. Even more importantly, only in the appendix one can read why the stiffness tensor can be represented as a matrix, and how the positive definiteness of a matrix can be ensured.
- On the other hand, background and related work are a bit repetitive
- Code / or pseudocode would be pretty helpful to understand output layers and how the stiffness tensor is ensembled.
- There are no runtime comparisons in the paper, especially since e3nn based models are known to be slow this would be interesting to know.
- Related to runtime comparisons, comparisons to FE methods are needed to get a better understanding of the presented performances. I understand that FE models are used as ground truth and that this might be tricky to obtain, but for example one could estimate runtime comparisons (especially since it is stated the FE methods are very slow) and report performance differences for nodal perturbations to get a perspective for the reported loss values? The latter should give a feeling to what nodal perturbations the presented losses are comparable.
- As far as I can see there is no OOD experiments although this is claimed to be one of the main reasons why such models built on physical principles are built.

**Questions:**

- When exactly is the positive semi-definite layer applied? after the readout?
- In Figure 2 validation loss of CGC+tr seems to be still dropping sharply, is it possible that training was stopped too early?

---

> ### Author Response · Authors · 2023-11-18
> **Response to Reviewer kbo4**
>
> We thank the reviewer for the helpful comments.
> We strive to improve the quality of our submission and we addressed all of the reviewer's comments.
>
> Specifically, we have made the following revisions in response to the reviewer's comments
>
>
> 1.  The reviewer states that
>
> > most of the information which is novel to the deep learning community is put into the appendix.
> > [...] the background and related work are a bit repetitive
>
> We expand on the solid mechanics section in the background, noting that for energy conservarion, the stiffness tensor must be positive semi-definite. We moved the details of MACE model which might feel repetitive into the appendix.
>
> 2.  The reviewer states that
>
> > Code / or pseudocode would be pretty helpful to understand output layers and how the stiffness tensor is assembled.
> > [...] When exactly is the positive semi-definite layer applied? After the readout?
>
> In the Methods section, we included a schematic of the model, so it is clear where the PSD layer is applied.
> We expanded the Methods section to include more details about the dataset and the positive semi-definite (PSD) layer.
> We explain how the fourth-order tensor is represented by $6\times6$ matrix and how it can be made positive definite.
> In addition, we have written a mathematical proof that our PSD layer in Mandel notation is equivariant. This is now in the Appendix.
>
> 3.  The reviewer points out that
>
> > There are no runtime comparisons in the paper
> > [...] comparisons to FE methods are needed [...]
>
> We provide inference time comparisons for the three families of ML models and FE baseline.
> While the e3nn-based model is slower than the ML counterparts, all ML models are approximately 3 orders of magnitude faster than FE.
>
> 4.  With regards to OOD experiments, we now explain in the Methods section (subset Dataset) the big separation of our training and test data, so testing is done on OOD data.
>
> 5.  While it appears in Figure 2 (now Fig 3) that validation loss for CGC+tr continues to drop sharply, this is exaggerated by the log-scale nature of the x-axis.
> All models were trained to convergence.

---

> > ### Comment · Reviewer_kbo4 · 2023-11-21
> > **Post-rebuttal**
> >
> > Thanks to the authors for restructuring the paper, many things have become clearer to me. I do thing that this work is an important contribution to the community, I have thus raised my score.

---

### Official Review · Reviewer_azcZ · 2023-11-01

**Soundness:** 2 fair
**Presentation:** 2 fair
**Contribution:** 2 fair
**Rating:** 5
**Confidence:** 4

**Summary:**

This paper proposes an application of equivariant GNNs for predicting the stiffness tensor of architected lattice metamaterials. To ensure the validity of physical significance, a layer is proposed to preserve the positive semi-definiteness of the predicted stiffness.

**Strengths:**

1.	There is less work focused on studying higher-order tensors beyond first-order tensors, such as coordinates, velocity, and force. Research on a 4th-order tensor represents a new situation for application.
2.	This paper try to design a new module to ensure the validity of physical significance, which is inspiring.
3.	The figures are pretty.

**Weaknesses:**

1.	This work chooses to predict a 4-th ordered tensor – “stiffness tensor”, whose symmetry has been maintain well with existing models like TFN, PAINN. This paper does not propose theoretical innovations in maintaining equivariance, so this methodology and the emphasis on “fourth-order” in abstract are not directly related (the Positive Semi-Definite Layer is based on physical meaning, regardless of whether it is a fourth-order tensor).
2.	Since the whole backbone is built with existing MACE layer, the main novelty is to propose the  “Positive Semi-Definite Layer”(let us call it “PSD-layer”). About this, here are some questions:
     - a) To maintain the equivariance of the total model, we must ensure each layer in such a model is equivariant. But for the PSD-layer, the PSD matrix $A$ is based on $M$, which is created by arrange $n(n+1)/2$ entries of the output of an equivariant model.
        - **i**. Is it an equivariant operator to arrange $n(n+1)/2$ entries to get $M$?  This requires a formal mathematical proof.
        - **ii**. Is it an equivalent operator of the function $\rho$ to turn $M$ into $A$? In the end of Section A.2.2, “If the overall model had been equivariant with respect to vectors in $U$, it will remain equivariant after eigenvalues are made positive” is not trivail, it may also require a formal mathematical proof.
         - **iii**.	For now, we assume that the previous question (function $\rho$ is equivariant) has a good proof. It is necessary to discuss whether this design will reduce the representational capacity of the entire neural network. Assume that this neural network is a bijection (we consider an equivalence class divided based on different perspectives as an input. But the function $\rho$ may not a bijection, e.g. $(\pm \Lambda) ^2$ will get a same result. The Wigner-D based network build a faith representation of the group, but the induction of function $rho$ will lead to an unfaith representation, it requires more analysis. For example,  an analysis for the increased stiffness-based errors of CGC+ve, NNConv+ve, and MACE+ve in Table 1 may be a manifestation of the decline in the ability of network representation.
    - b)	Let assume the PSD-layer is equivariant, there are some questions about the experiments:
        - **i**.	In Section 3 (Related Work) and Section 5(Conclusion), Finite element (FE) modelling is mentioned, but the experiment could not find the data comparison (e.g. time cost or accuracy) between your model and FE.
        - **ii**.	More powerful baselines should be discussed, especially the geometric GNNs like TFN[1], NequIP [2], SCN[3], GMN[4], SE(3)-Transformer [5].
3.	Here are some possible typos and recommended symbol modifications:
    - a)	Almost all the inline formulas in the article lack punctuation at the end.
    - b)	In Section 2.1, the rotation of the stiffness tensor $R_{ia}R_{jb}C_{abcd}R_{kc}R_{ld}$ may be better to generalize to $n$-th order tensors with the format of changing bases of a tensor($ R_{ia}R_{jb} R_{kc}R_{ld} C_{abcd}$).
    - c)	In Eq. (2),  the Wigner-D matrix generally written in the form $D_{m’,m}^{(l)}(R)$.
    - d)	In Section 4, the third line in the paragraph “Positive Semi-Definite Layer”, “in line wih”, may be “in line with”.

Reference:

- [1] Tensor field networks: Rotation-and translation-equivariant neural networks for 3d point clouds.
- [2] E(3)-equivariant graph neural networks for data-efficient and accurate interatomic potentials
- [3] Spherical Channels for Modeling Atomic Interactions
- [4] Equivariant Graph Mechanics Networks with Constraints
- [5] SE(3)-Transformers: 3D Roto-Translation Equivariant Attention Networks

**Questions:**

Please refer to the weakness.

---

> ### Author Response · Authors · 2023-11-18
> **Response to Reviewer azcZ**
>
> We would like to thank the reviewer for the detailed comments.
> It is clear that the reviewer wants to ensure the quality of the submission; thus, we have made significant changes to address all of the reviewer's comments.
>
> ### Comment 1
> The reviewer states that
>
> >    the existing models (like TFN and PAINN) have been able to fit a 4-th order tensor and that this paper does not propose theoretical innovations in maintaining equivariance.
>
>
> While it is true that models like TFN and PAINN can fit 4-th-order tensors, there is a lack of such examples in the machine learning community.
> For instance, the PAINN paper shows an example of fitting rank-2 polarizability tensor.[1,p5]
> Very few, if any, examples of rank-4 tensors have been reported in the machine learning community.
> We believe that it is one of the contributions of this paper (in the area of *applications to physical sciences*) to make available a real-world dataset of rank-4 tensors, as this could fuel further developments in the community.
>
> Secondly, Positive Semi-Definite (PSD) layer brings about new challenges related to equivariance.
> Maintaining equivariance of the 4-th order tensor is more challenging than for the rank-2 tensor.
> A new theoretical contribution in maintaining equivariance, consistent with positive-definiteness of a 4-th order tensor, is needed and is presented here.
>
> ### Comment 2
> The reviewer raises valid questions. Based on these, we have made the following revisions to the manuscript.
>
>
> 1.  We included in the Appendix section **Proof of equivariance of PSD layer in Mandel notation**
> in which we mathematically prove that our PSD layer is equivariant.
> 1.  The reviewer voices concerns that function $\rho$ might lead to a reduced representational capacity of the model.
> While we are unable to run further experiments at this short timescale, we will make effort to address reviewer's points in due course.
> Empirically and from the point of view of using this model in practice, the performance hit due to the possible reduction in representational capacity is minor as seen in Table 1.
> 1.  We have included section **Speedup using machine learning methods** in the results section
> where we compare inference speed of the three families of models and FE.
> We show that the ML models bring a speedup of 3 orders of magnitude.
> Since FE results are the ground truth data for ML training, it is not meaningful to compare the accuracy of FE with the accuracy of ML models.
> 1.  The reasoning behind our decision to discuss models based on CGC, NNConv, and MACE is as follows.
> The only GNN models, to our best knowledge, that have been used for lattices are CGC and NNConv. [2,3]
> Therefore we decided to benchmark our model against the previous baseline used in the literature.
> The reason why we chose MACE as the basis for our model (as opposed to TFN, NequIP etc) is that
> it has been reported by the computational chemistry community as the best equivariant model for molecules.
> As a side note, in paragraph **Correlation order** on p.9 we note that MACE with $\nu=1$ is equivalent to TFN.
>
>
> ### Comment 3
> The reviewer makes valuable suggestions and we have revised the manuscript based on their comments.
>
> Reference:
> [1] Equivariant message passing for the prediction of tensorial properties and molecular spectra
> [2] Using Graph Neural Networks to Approximate Mechanical Response on 3D Lattice Structures
> [3] Graph-based metamaterials: Deep learning of structure-property relations

---

> > ### Comment · Reviewer_azcZ · 2023-11-21
> > **Received with thanks.**
> >
> > Dear author,
> >     I've read your responses. Some concerns, such as comparison with FE and Proof of equivariance of PSD have been addressed. However, I still have the concerns about the  theoretical innovations, mathematical proof of $\phi$ and comparisons with other geometric GNNs. For now, I will maintain my negative opinions and finalize my score after the reviewers' discussion.

---

### Author Response · Authors · 2023-11-18
**General response to reviews**

We thank the referees for their constructive comments.
The main concern raised by the referees is that the paper presents a relatively minor enhancement to the MACE architecture.
We recognise this but emphasize that the key contribution of the paper lies in the application to a physical science problem, viz. the first attempt at creating a dataset for a 4th order tensor (the elasticity tensor) \& modifying MACE for this problem.
This opens the field to the wide application of ML to structural mechanics.
In this regard our specific contributions are


(a)  The proposed PSD layer -- as reviewers azcZ and kbo4 point out, ensuring the validity of physical significance is inspiring, and it's a new tool for physics-based machine learning. Moreover, as reviewer azcZ suggests by asking for a formal mathematical proof, it is not trivial to ensure the equivariance of the PSD layer.
We now provide this proof.

(b)  Creating and releasing the dataset, and modifying the MACE architecture to predict the 4th order elasticity tensor of metamaterials.
This task had not been accomplished prior to this work.

We believe that we do provide justifiable contribution in the area of *applications to physical sciences* and we thank you for considering our submission.
We address all the specific points raised by the individual referees below and in the updated version of the manuscript.

---

> ### Author Response · Authors · 2023-11-21
> **General response to reviews 2.0**
>
> We thank the reviewers for iterating on their feedback and helping us improve the quality of the submission.
> Based on the second round of comments, we revised the paper.
> In the newly updated version, we
>
> (i) refine based on comments by reviewer uV3y -- we take care that definitions are explained when they first appear, and we are more precise with respect to the introduction of FEM;
>
> (ii) include an example of downstream application as suggested by reviewer TDG7.

---

### Meta-Review · Area_Chair_1MNo · 2023-12-11

**Metareview:**

This paper presents a GNN approach with higher-order tensors and show how it can predict useful elasticity properties of materials. The reviewers agree this is an interesting approach that is well-motivated by the application, and that the application domain is very meaningful. While there is slight disagreement between the reviewers, overall there is consensus on the value of this paper, and I recommend acceptance.

**Justification For Why Not Higher Score:**

n/a since it is borderline accept

**Justification For Why Not Lower Score:**

n/a

---

### Decision · Program_Chairs · 2024-01-16

Accept (poster)